# Metronomic chemotherapy offsets HIFα induction upon maximum-tolerated dose in metastatic cancers

Luana Schito[1,2,3,*,†] (iD), Sergio Rey[1,2,†], Ping Xu[3], Shan Man[3], William Cruz-Muñoz[3] & Robert S Kerbel[3,4,**] (iD)

## Abstract

Conventional maximum-tolerated dose (MTD) chemotherapy relies on periodic, massive cancer cell ablation events followed by treatment-free intermissions, stereotypically resulting in resistance, relapse, and mortality. Furthermore, MTD chemotherapy can promote metastatic dissemination via activation of a transcriptional program dependent on hypoxia-inducible factor (HIF)-1α and (HIF)-2α (hereafter referred to as HIFα). Instead, frequent low-dose metronomic (LDM) chemotherapy displays less adverse effects while preserving significant pre-clinical anticancer activity. Consequently, we hereby compared the effect of MTD or LDM chemotherapy upon HIFα in models of advanced, metastatic colon and breast cancer. Our results revealed that LDM chemotherapy could offset paralog-specific, MTD-dependent HIFα induction in colon cancers disseminating to the liver and lungs, while limiting HIFα and hypoxia in breast cancer lung metastases. Moreover, we assessed the translational significance of HIFα activity in colorectal and breast TCGA/microarray data, by developing two compact, 11-gene transcriptomic signatures allowing the stratification/identification of patients likely to benefit from LDM and/or HIFα-targeting therapies. Altogether, these results suggest LDM chemotherapy as a potential maintenance strategy to stave off HIFα induction within the intra-metastatic tumor microenvironment.

**Keywords** breast cancer; colon cancer; HIF-1; hypoxia; low-dose metronomic
**Subject Category** Cancer

## Introduction

Hypoxia (low $O_2$) is a pervasive microenvironmental feature of solid cancers, associated with disease progression and poor survival due to metastatic dissemination and resistance to cancer therapies (Schito & Semenza, 2016; Rey et al, 2017; Schito, 2019). Mechanistically, hypoxia is transduced to the *nucleus* of cancer cells by the activation of a transcriptional program mediated by hypoxia-inducible factor (HIF)-1α and (HIF)-2α (henceforth referred to as HIFα) (Schito & Semenza, 2016). In particular, both hypoxia and HIFα have been linked to therapy resistance, metastatic progression, and mortality in a variety of cancers, including colon and breast carcinomas, wherein the efficacy of conventional maximum-tolerated dose (MTD) chemotherapy is rather limited in late-stage, metastatic disease (Shimomura et al, 2013; Dekervel et al, 2014; Schito & Rey, 2017), a condition often managed via maintenance chemotherapy. We have previously suggested that MTD chemotherapy could aggravate intra-tumoral hypoxia through HIFα-dependent mechanisms counteracting cancer cell killing (Rey et al, 2017). In this context, we hypothesized that low-dose metronomic (LDM) chemotherapy, a modality with advantageous safety, tolerability, and possibly therapeutic profiles when administered as maintenance, might serve as a tool to offset intra-tumoral HIFα levels caused by conventional MTD chemotherapy (Cao et al, 2013; Samanta et al, 2014; Simkens et al, 2015; Kerbel & Shaked, 2017; Bisogno et al, 2018). In order to test this hypothesis, we performed the first side-by-side comparison of oral LDM cyclophosphamide + capecitabine, a doublet regimen previously evaluated in advanced breast cancers (Dellapasqua et al, 2008), as opposed to an equivalent MTD regimen. Further, we implemented automated HIFα quantification algorithms in orthotopic models of advanced colon and breast cancers that reproducibly metastasize to the liver and lungs (Teicher et al, 1990; Hackl et al, 2013; Shaked et al, 2016). Our data show that LDM chemotherapy can offset paralog-specific, MTD-triggered HIFα induction in colon adenocarcinomas disseminating to the liver and

1  UCD School of Medicine, University College Dublin, Dublin 4, Ireland
2  UCD Conway Institute of Biomolecular & Biomedical Research, University College Dublin, Belfield, Dublin 4, Ireland
3  Biological Sciences Platform, Sunnybrook Research Institute, Toronto, ON, Canada
4  Department of Medical Biophysics, University of Toronto, Toronto, ON, Canada
   *Corresponding author. Tel: +1 (416) 480 5711; E-mail: luana.schito@ucd.ie
   **Corresponding author. Tel: +353 (1) 716 6700; E-mail: robert.kerbel@sri.utoronto.ca
   †These authors contributed equally to this work

lungs, while limiting intra-metastatic hypoxia in breast cancer lung nodules. In addition, we explored the translational potential of these pre-clinical findings by utilizing statistical modeling and bioinformatics to generate two compact HIFα gene signatures designed to uncover the prognostic consequences of HIFα activation in colon and breast cancer patients. The data herein presented suggest that maintenance LDM chemotherapy may improve overall chemotherapy outcomes by offsetting HIFα upregulation upon conventional MTD regimens, while providing two novel transcriptomic tools allowing the identification of cancer patients with high HIFα transactivity, and thus potentially amenable to combinatorial LDM and/ or HIFα targeting.

# Results

### LDM chemotherapy offsets HIF-1α induction in colon cancers

Quantitative analysis of orthotopic human (HT29) primary colonic adenocarcinomas showed that doublet LDM cyclophosphamide + MTD capecitabine increased median nuclear HIF-1α protein levels by 12-fold compared to vehicle-treated controls ($P = 3.9 \times 10^{-3}$; Fig 1A and Appendix Fig S1A); by contrast, switching to a doublet LDM cyclophosphamide+capecitabine regimen blocked ≈97% of HIF-1α induction ($P = 2 \times 10^{-3}$; Fig 1A and Appendix Fig S1A). Notably, both nuclear and cytoplasmic HIF-2α levels were unaffected by chemotherapy (Fig 1B and Appendix Fig S1B), thus suggesting a paralog-specific effect. We did not observe chemotherapy-dependent induction of CA9, a known HIF-1α target commonly utilized as a *proxy* molecular readout for hypoxia and HIFα activity (Fig 1C). These data suggest cell-autonomous effects and the need for caution whenever CA9 is used as a substitute for HIFα activity in hypoxic cancers. Furthermore, HIF-1α induction by doublet LDM cyclophosphamide + MTD capecitabine was not due to differences in tumor volume, since luminescence plots displayed comparable slopes and absolute values at endpoint (Appendix Fig S2A), overall signal doubling times of ≈15 days (Appendix Fig S2B), and comparable cross-sectional areas among treatments (Appendix Fig S2C and D); likewise, no significant correlations were observed among HIF-1α, HIF-2α and CA9 expression (Appendix Fig S3A–C).

Since HIF-1α activation triggers cell cycle arrest *in vitro* (Hubbi *et al*, 2013), we functionally validated these *in vivo* data using machine-learning algorithms measuring Ki67-dependent proliferation. Our results uncovered an inverse correlation between HIF-1α and nuclear Ki67 (Fig 1D), an effect that was not observed for HIF-2α or CA9 (Appendix Fig S3D and E). Of note, we did not observe significant effects exerted by MTD or LDM chemotherapy regimens upon Ki67 *per se* (Fig 1D), although the mean proliferative index in all tumors was 29.2% [range: 7.6–89.5%; $n = 29$], consistent with our observations of similar primary tumor volumes, thereby expected to result in similar $O_2$ diffusion distances and hence hypoxic fractions.

### LDM chemotherapy offsets HIF-1α induction in colonic metastases to the liver

At endpoint, liver metastatic nodule diameters derived from primary colonic tumors followed a log-normal distribution (Fig 2A). Using

the median diameter of vehicle-treated metastases (970 μm) as a reference, only doublet LDM cyclophosphamide+capecitabine treatment induced a significant decrease in colon-derived metastatic liver nodule size by automated morphometric analysis ($P = 0.0299$ by chi-square test; Fig 2A).

We next assessed the effect of MTD and LDM doublet chemotherapy on HIFα in the same advanced metastatic, preclinical colon cancer setting. Consistent with primary tumor data, doublet LDM cyclophosphamide + MTD capecitabine increased HIF-1α in liver metastases by 2.5- fold ($P = 3.1 \times 10^{-2}$; Fig 2B and Appendix Fig S4A), an induction that was blunted by doublet LDM cyclophosphamide+capecitabine (≈95% decrease, $P = 8 \times 10^{-4}$), equivalent to a reduction of ≈ 87% as compared to vehicle-treated controls ($P = 5.3 \times 10^{-3}$; Fig 2B and Appendix Fig S4A). These data suggest that LDM chemotherapy can attenuate HIF-1α induction in liver metastases independently of diffusion-limited hypoxia and in the presence of equivalent metastatic burdens. By contrast, MTD and LDM chemotherapy schemes did not modulate liver metastatic HIF-2α levels (Appendix Fig S4B and C), wherein expression was predominantly cytoplasmic (Appendix Fig S4D). In addition, HIF-1α and HIF-2α expression within liver metastases or their peri-metastatic parenchyma was not correlated (Appendix Fig S4E). Notwithstanding, automated quantitative analysis of the non-malignant peri-metastatic rim of colon metastases to the liver [median thickness: 213 μm (range: 49–353 μm)] revealed a striking log-linear correlation, wherein metastatic HIF-1α or -2α levels predicted paralog expression in the surrounding parenchyma, therefore suggesting a hitherto unrecognized effect of the intra-metastatic environment upon surrounding, non-malignant hepatocytes (Fig 2C and D).

### LDM chemotherapy offsets HIFα induction in colon and breast cancer metastases to the lung

Not unlike their clinical counterparts, experimental colon cancers disseminated to the lungs, wherein nodule size distribution was log-normal, similarly to liver metastases, albeit presenting significantly smaller nodular diameters (Fig EV1A). The median cross-sectional diameter of colon cancer lung metastases in vehicle-treated controls was ≈175 μm; remarkably, metastatic diameter was significantly decreased by LDM monotherapy with capecitabine (≈44%) or cyclophosphamide (≈53%) as compared to vehicle controls ($P < 0.05$ by one-way ANOVA and chi-square tests; Fig EV1A). By contrast, MTD capecitabine increased lung nodule sizes by ≈1.5-fold versus vehicle ($P = 0.012$; Fig EV1A). Furthermore, all chemotherapy combinations, except for LDM capecitabine, significantly increased HIF-2α expression by various degrees ($P < 0.05$ versus vehicle; Fig EV1B and Appendix Fig S5A). In particular, doublet LDM cyclophosphamide + MTD capecitabine increased HIF-2α expression by 33.4-fold ($P < 10^{-4}$; Fig EV1B), an effect that was blunted by ≈81% in the doublet LDM cyclophosphamide+capecitabine group ($P = 0.044$; Fig EV1B). Importantly, intra-metastatic HIF-2α expression increased as a function of nodule size (Fig EV1C), thereby suggesting dependency upon diffusion-limited hypoxia. Contrary to the liver microenvironment, HIF-1α expression in lung metastases was not significantly modulated by chemotherapy (Fig EV1D and Appendix Fig S5B), nonetheless revealing a positive correlation with nodular diameter, similarly to

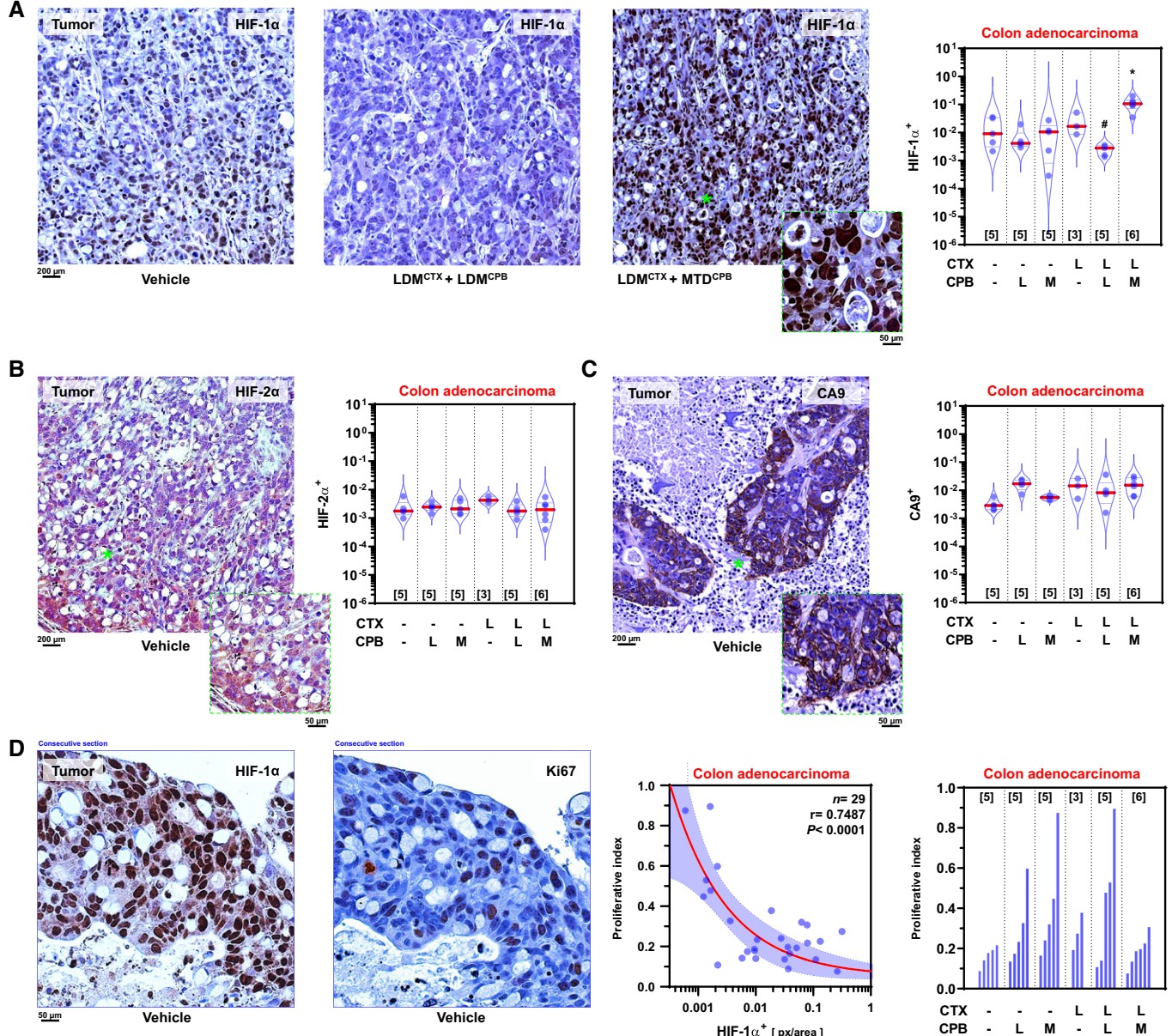

**Figure 1.  LDM chemotherapy selectively offsets HIF-1α levels in experimental colon cancers.**

A   HIF-1α levels in HT29 primary tumors. *Left*: vehicle-treated controls. *Middle/left*: Doublet LDM cyclophosphamide + capecitabine (LDM$^{CTX}$ + LDM$^{CPB}$). *Middle/right*: LDM cyclophosphamide + MTD capecitabine (LDM$^{CTX}$ + MTD$^{CPB}$). *Inset*, high-magnification image of the region marked with a green asterisk (*). *Right*: Quantification of the effect of monotherapies or doublet LDM/MTD regimens on HIF-1α$^+$ area. $F_{5,12}$ = 8.791 and $P$ = 0.001 for overall treatment by Brown–Forsythe ANOVA; *$P$ = 0.046 versus vehicle; $^\#P$ < 0.0001 versus LDM$^{CTX}$ + MTD$^{CPB}$ by Benjamini, Krieger, and Yekutieli *post hoc* test.

B   HIF-2α levels in HT29 primary tumors. *Left*: Example of immunostaining in vehicle-treated controls. *Inset*, high-magnification image of the region marked with a green asterisk (*). *Right*: Quantification of the effect of monotherapies or doublet LDM/MTD regimens on HIF-2α$^+$ area. $F_{5,18}$ = 1.215 and $P$ = 0.3424 (not significant) for overall treatment by Brown–Forsythe ANOVA.

C   CA9 levels in HT29 primary tumors. *Left*: Example of immunostaining in vehicle-treated controls. *Inset*, high-magnification image of the region marked with a green asterisk (*). *Right*: Quantification of the effect of monotherapies or doublet LDM/MTD regimens on CA9$^+$ area. $F_{5,11}$ = 2.466 and $P$ = 0.0961 (not significant) for overall treatment by Brown–Forsythe ANOVA.

D   Correlation between HIF-1α levels and proliferation indexes in HT29 primary tumors. *Left* and *middle/left*: HIF-1α and Ki67 expression in vehicle-treated controls; consecutive sections are shown. *Middle/right*: HIF-1α$^+$ versus Ki67$^+$ proliferation index scatterplot. Each point represents median values for HIF-1α$^+$ tumors. Regression line (red) and 95% CI (shaded blue area) are indicated. $F_{1,27}$ = 34.45 and $P$ = 10$^{-4}$; slope ≠ 0 by $F$-test. *Right*: Machine-learning quantification of the effect of monotherapies or doublet LDM/MTD regimens upon Ki67$^+$ proliferative index. Median indexes per tumor are indicated. $F_{5,12}$ = 1.609 and $P$ = 0.2312 (not significant) for overall treatment by Brown–Forsythe ANOVA.

Data information: Violin plots present 50$^{th}$ (red line), 25$^{th}$ and 75$^{th}$ percentiles (blue line); numbers in brackets indicate number of tumors per group. *L*, low-dose metronomic; *M*, maximum-tolerated dose; *r*, correlation coefficient. Low power magnification images of all experimental conditions can be found in Appendix Fig S1. Blue frames in D indicate consecutive sections stained for HIF-1α and Ki67.

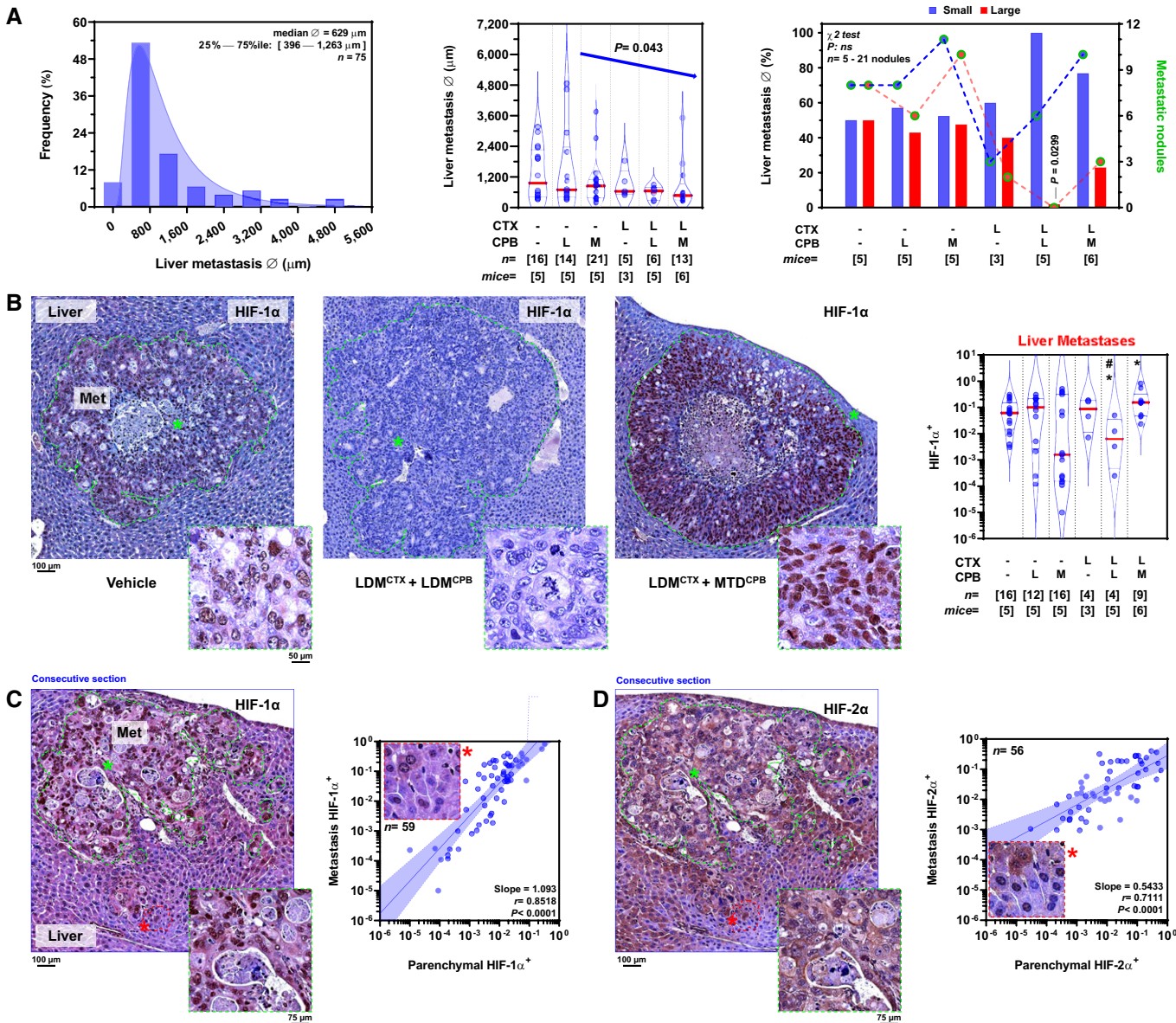

**Figure 2.  LDM chemotherapy selectively offsets HIF-1α levels in colon cancer metastases to the liver.**

A  Effect of LDM and MTD chemotherapy on HT29 liver metastatic nodule size.  *Left*: Histogram of pooled cross-sectional metastatic diameter (Ø). *Middle*: Metastatic diameter (Ø) by chemotherapy regimen. A decreasing linear trend for median metastatic diameter was observed (left to right, slope = −143.2, P = 0.043 by post-test for trend). *Right*: Dichotomized metastatic size at median diameter of vehicle-treated tumors. Liver nodule size was classified as small (blue) or large (red). Individual nodule counts per group and size category (small/large) are plotted on the right ordinate. $\chi^2$ (df = 5) = 6.80; P = 0.2361 (not significant) for overall effects on size; $\chi^2$ (df = 1) = 4.71; P = 0.0299 (LDM$^{CTX}$ + LDM$^{CPB}$ versus vehicle).

B  HIF-1α levels in HT29 liver metastatic nodules. *Left*: Vehicle-treated controls. *Middle/left*: Doublet LDM cyclophosphamide + capecitabine (LDM$^{CTX}$ + LDM$^{CPB}$). *Middle/right*: LDM cyclophosphamide + MTD capecitabine (LDM$^{CTX}$ + MTD$^{CPB}$). *Right*: Automatic quantification of the effect of monotherapies or doublet regimens on HIF-1α$^+$ areas in individual metastatic nodules. $F_{2,13}$ = 4.796 and P = 0.028 for overall treatment by Brown–Forsythe ANOVA; *P = 0.038 LDM$^{CTX}$ + MTD$^{CPB}$ versus vehicle or P = 0.0123 LDM$^{CTX}$ + LDM$^{CPB}$ versus vehicle; #P = 0.0098 LDM$^{CTX}$ + LDM$^{CPB}$ versus LDM$^{CTX}$ + MTD$^{CPB}$ by Benjamini, Krieger, and Yekutieli *post hoc* test.

C  Correlation between metastatic and peri-metastatic HIF-1α levels in the liver. *Left*: Nodule immunostaining and parenchymal HIF-1α expression. Each point represents median values per nodule. Regression line (blue) and 95% CI (shaded blue area) are indicated. $F_{1,57}$ = 150.7 and P < 0.0001; slope ≠ 0 by F-test. *Right*: Correlation between intra-metastatic and peri-metastatic parenchymatous ring in the liver. Each point represents median values per nodule. Regression line (blue) and 95% CI (shaded blue area) are indicated. $F_{1,54}$ = 55.2 and P < 0.0001; slope ≠ 0 by F-test.

D  Correlation between metastatic and peri-metastatic HIF-2α levels in the liver. *Left*: Nodule immunostaining and parenchymal HIF-2α expression. *Right*: Correlation between intra-metastatic and peri-metastatic parenchymatous ring in the liver. Each point represents median values for each nodule. Regression line (blue) and 95% CI (shaded blue area) are indicated. $F_{1,54}$ = 55.2 and P < 0.0001; slope ≠ 0 by F-test. n, number of nodules.

Data information: Violin plots present 50th (red line), 25th, and 75th percentiles (blue line); numbers in brackets indicate number of nodules (n) or animals (mice). L, low-dose metronomic; M, maximum-tolerated dose; ns, not significant; Met, metastasis; r, correlation coefficient. Dashed green lines encircle the histological limit between metastatic nodules and their surrounding normal liver parenchyma. Insets show high-magnification images of regions marked with asterisks (*). Blue frames in C and D indicate consecutive sections from the same liver metastasis, stained for HIF-1α or HIF-2α, respectively.

HIF-2α (Appendix Fig S5C). Since intra-metastatic HIF-1α and HIF-2α levels were not correlated in the lung (Appendix Fig S5D), we suggest that these data indicate a site-specific effect wherein MTD chemotherapy selectively stimulates HIFα paralog induction in experimental colon adenocarcinomas (i.e., HIF-1α: liver and HIF-2α: lung), an effect that was selectively offset by specific LDM regimens.

In light of these results, where colonic metastases to the liver were ≈3.6-fold larger than in the lung, we aimed to determine whether paralog specificity was due to microenvironmental factors or rather a consequence of differences in hypoxic levels, dependent on metastatic size, burden, and/or factors related to the physiology of the host organ. Likewise, we sought to rule out putative systemic perfusion effects due to chemotherapy by quantifying fluorescently labeled dextran injected at tissue harvesting, while determining intra-metastatic hypoxic fractions through pimonidazole labeling. These experiments were carried out in a cisplatin-resistant, immunocompetent metastatic breast cancer model (EMT6-CDDP) that disseminates to the lungs, wherein LDM chemotherapy has been shown to prolong survival (Shaked *et al*, 2016). In this immunocompetent, syngeneic model, mice received 9 days of adjuvant MTD and LDM chemotherapy after primary breast tumor resection that was programmed to occur before the onset of mortality due to overt metastatic disease. These experiments revealed lung nodules in > 50% of mice (median diameter = 626 μm; Fig 3A) of almost identical size to liver metastases in the colon adenocarcinoma model (median diameter = 629 μm; Fig 2A). Remarkably, intra-metastatic hypoxic fractions were significantly decreased by both LDM monotherapies (≈77% in LDM capecitabine and ≈ 82% in LDM cyclophosphamide, $P = 1.9 \times 10^{-3}$; Fig 3B). Similarly, doublet LDM cyclophosphamide+capecitabine decreased intra-metastatic hypoxia by ≈73% when compared with doublet LDM cyclophosphamide + MTD capecitabine ($P = 1.1 \times 10^{-2}$; Fig 3B). In parallel, HIF-1α expression was significantly decreased by ≈97%, ≈94%, or ≈89% in LDM capecitabine, LDM cyclophosphamide, or doublet LDM cyclophosphamide+capecitabine, respectively ($P < 0.01$; Fig 3C). Furthermore, microvessel density, as measured by CD31 immunoreactivity, reached a minimum within the LDM cyclophosphamide monotherapy group (≈65% reduction versus vehicle, $P = 4 \times 10^{-4}$; Fig 3D). In addition, quantitative analysis of metastatic lung nodules revealed a positive correlation among hypoxic fractions, microvessel density (Fig EV2A), and HIF-1α (Fig EV2B). Likewise, intra-metastatic HIF-1α expression positively correlated with microvessel density (Fig EV2C). Altogether, these data suggest that intra-tumoral hypoxia exerts a central role upon HIF-1α-dependent vascularization in metastatic breast cancers disseminated to the lungs. Importantly, estimates of lung perfusion in this model, utilizing intravascular fluorescently labeled dextran, did not show differences among chemotherapeutic regimes (Appendix Fig S6A and B), suggesting that intra-metastatic hypoxia in the lung does not depend on systemic effects and it is rather a local microenvironmental effect (Appendix Fig S6C and D). Furthermore, these experiments suggest that HIFα paralog induction in metastatic nodules depends on diffusion-limiting hypoxia, since identically sized liver or lung metastases from colon or breast adenocarcinomas both expressed HIF-1α, an induction that was offset by LDM chemotherapies. In addition, we observed a negative correlation between Ki67 proliferative indexes and HIF-1α expression in breast adenocarcinoma metastases to the lung (Appendix Fig

S7), in line with our findings in experimental colon adenocarcinomas (Fig 1D). Therefore, these observations provide *in vivo* evidence supporting HIF-2α as a promoter of early metastatic colonization through proliferation in incipient metastatic lesions (as seen in HT29 lung metastases); by contrast, HIF-1α could play a counterbalancing role in larger secondary tumors (as seen in HT29 liver and lung EMT6-CDDP metastases), by promoting cell cycle arrest as a protective mechanism against chemotherapy-induced tumoral ablation, in line with previously published *in vitro* work (Gordan *et al*, 2007; Hubbi *et al*, 2013).

### HIFα activity predicts mortality and metastasis in colon and breast cancer patients

To assess the potential clinical significance of increased HIFα expression in MTD and doublet LDM chemotherapy, we analyzed publicly available transcriptomic colon and breast cancer data from TCGA and GEO datasets. A mechanistically informed search for experimentally validated HIFα targets, extracted from a previously published, predictive gene signature in colon cancer patients (Dekervel *et al*, 2014), revealed a "core" of 38 (colon) or 29 (breast) HIFα-inducible transcripts. *Spearman* regression matrices of both cancer types revealed that > 70% of possible transcript combinations were statistically significant after Bonferroni *post hoc* correction ($\alpha < 10^{-5} = P < 0.01$; Fig 4A). Notwithstanding, in light of recent data indicating that > 90% of random gene signatures containing > 100 transcripts can predict cancer survival *per se* (Venet *et al*, 2011), we aimed at decreasing the number of transcripts to a minimum. Unbiased statistical modeling yielded two novel compact, eleven HIFα-inducible transcript signatures (hereafter referred to as HIFα-inducible (HIFi) colon cancer score [HIFi-CCS; Fig 4B] and breast cancer score [HIFi-BCS; Fig 4C]). Dichotomization of colon and breast TCGA data according to HIFi-CCS or HIFi-BCS at the median confirmed individual (> 1.5-fold) upregulation of 8/39 (21%) or 5/29 (17%) of the statistically significant HIFα "seeds" for colon and breast cancers, respectively (Fig 4D). Importantly, Kaplan–Meier analysis showed that patients with increased HIFα transcript levels, as detected by HIFi-CCS or HIFi-BCS, presented decreased overall and recurrence-free survival (Fig 4B and C). Furthermore, HIFi-CCS was higher in colon cancer patients with nodal invasion, wherein high scores were also associated with accelerated progression to metastasis [median: 11.4 versus 28.4 months in high versus low HIFi-CCS, respectively; Fig 4B]. Similarly, HIFi-BCS was higher in estrogen receptor-negative tumor-bearing patients, while predicting shortened distant-metastasis-free survival (Fig 4C).

## Discussion

Despite abundant data supporting the notion that HIF-1α can promote every single aspect of the multistep metastatic cascade (Schito & Semenza, 2016; Schito & Rey, 2017), only a few clinical studies have addressed HIFα expression in metastatic cancers and its role in disease progression (Cao *et al*, 2009; van der Wal *et al*, 2012; Shimomura *et al*, 2013). Therapeutically, despite the favorable tolerability and safety profiles of LDM chemotherapies, no side-

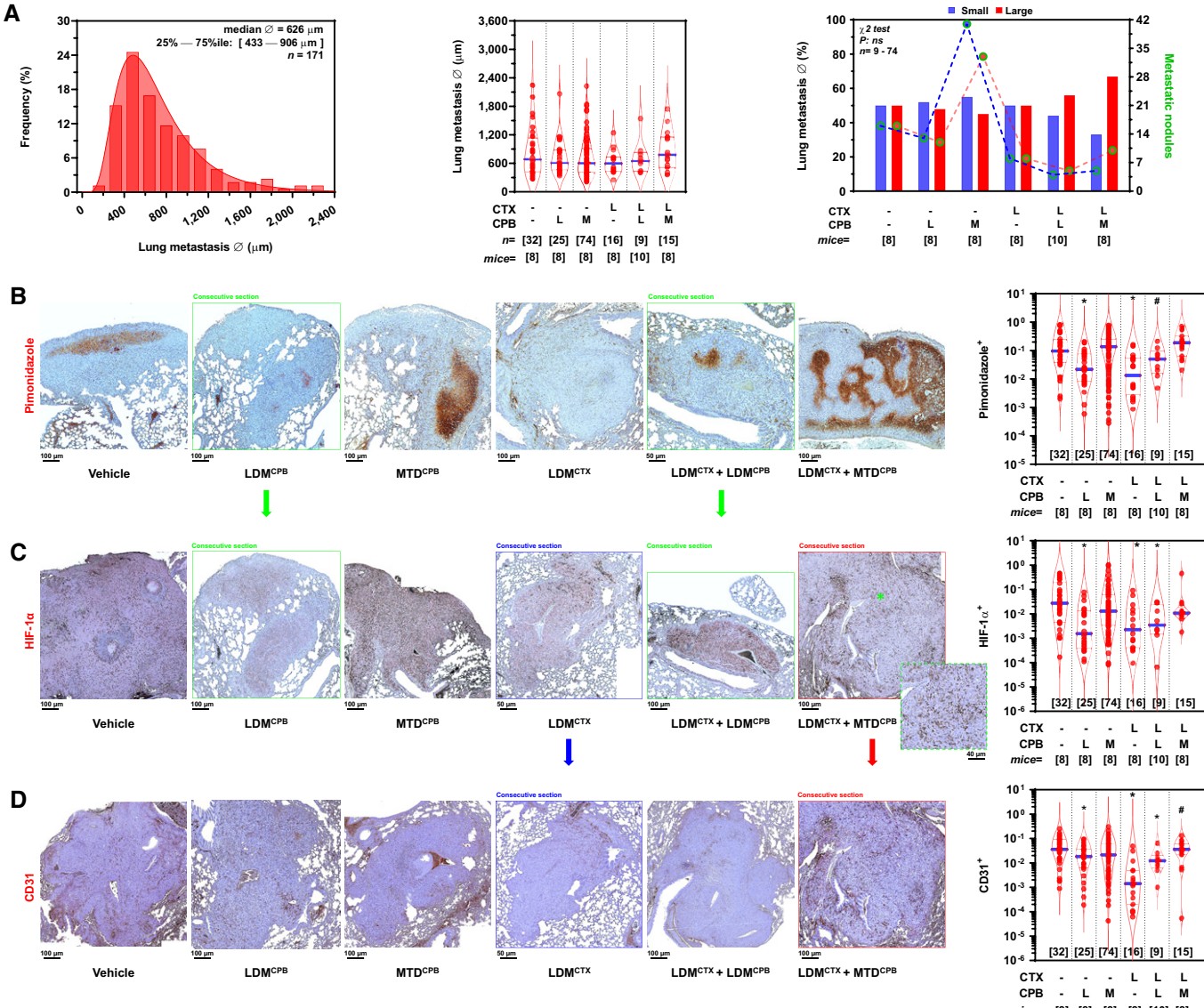

**Figure 3. LDM chemotherapy offsets HIF-1α levels in breast cancer metastases to the lung.**

A  EMT6-CDDP lung metastatic nodule size. *Left*: Histogram of cross-sectional metastatic diameter (∅). *Middle*: Metastatic diameter (∅) by chemotherapy regimen. $F_{5,99}$ = 0.9310 and $P$ = 0.4644 (not significant) for overall treatment by Brown–Forsythe ANOVA. *Right*: Dichotomized metastatic size at median diameter of vehicle-treated tumors. Lung nodule size was classified as small (blue) or large (red). Individual nodule counts per group and size category (small/large) are plotted on the right ordinate. $\chi^2$ (df = 5) = 2.631; $P$ = 0.7567 (not significant) for overall effects on size.

B  Hypoxic fraction in lung metastatic nodules. Pimonidazole adduct immunoreactivity was automatically quantified and expressed as fractional positive areas per nodule (*rightmost panel*). $F_{5,110}$ = 7.544 and $P$ < 0.0001 for overall treatment by Brown–Forsythe ANOVA; *$P$ < 0.01 versus vehicle; [#]$P$ = 0.0111 doublet LDM cyclophosphamide + capecitabine versus LDM cyclophosphamide + MTD capecitabine by Benjamini, Krieger, and Yekutieli *post hoc* test.

C  HIF-1α levels in lung metastatic nodules. Automatic quantification of HIF-1α levels measured as fractional positive areas per nodule (*rightmost panel*). $F_{5,114}$ = 5.325 and $P$ = 0.0004 for overall treatment by Brown–Forsythe ANOVA; *$P$ < 0.01 versus vehicle by Benjamini, Krieger, and Yekutieli *post hoc* test.

D  Microvessel density in lung metastatic nodules. Automatic quantification of CD31 fractional areas per nodule (*rightmost panel*). $F_{5,124}$ = 7.531 and $P$ < 0.0001 for overall treatment by Brown–Forsythe ANOVA; *$P$ < 0.01 (LDM capecitabine, cyclophosphamide, or doublet capecitabine + cyclophosphamide versus vehicle), [#]$P$ = 0.0417 (doublet LDM capecitabine + cyclophosphamide versus doublet LDM cyclophosphamide + MTD capecitabine) by Benjamini, Krieger, and Yekutieli *post hoc* test.

Data information: Violin plots present 50[th] (blue line), 25[th], and 75[th] percentiles (red line); numbers in brackets indicate number of nodules (*n*) or animals (mice) per group. *CPB*, capecitabine; *CTX*, cyclophosphamide; *L*, low-dose metronomic; *M*, maximum-tolerated dose. *Inset*: high-magnification image of the region marked with an asterisk (*; C *rightmost panel*). Green, blue, and red frames in B, C, and D indicate consecutive sections from the same lung metastatic nodule, stained for pimonidazole (hypoxia), HIF-1α, or CD31 (microvascular density).

by-side comparison of LDM and MTD therapeutic modalities upon HIFα has been carried out to date.

Analysis of primary and metastatic HIF-1α or HIF-2α levels in a total of ≈9,300 images across two different models of advanced cancer (colon and breast) revealed that MTD chemotherapy can selectively induce HIF-1α in primary tumors and established liver or lung metastases, an effect that was offset by LDM monotherapies and/or doublet regimens. HIF-2α upregulation by MTD regimens was less marked and only observed in incipient colonic metastases

to the lung, an effect that was tempered by LDM monotherapies with cyclophosphamide or capecitabine. Size differences among colorectal lung and liver metastases seem, in this context, likely to be due to asynchronic metastatic colonization, since the colon cancer model herein utilized disseminates primarily via the portal system into the liver, and secondarily, into the lung via the systemic circulation, mimicking colon cancers in the clinical setting. Importantly, our results in the second model of advanced breast cancer disseminating to the lungs, suggest selective upregulation of HIF-1α by MTD

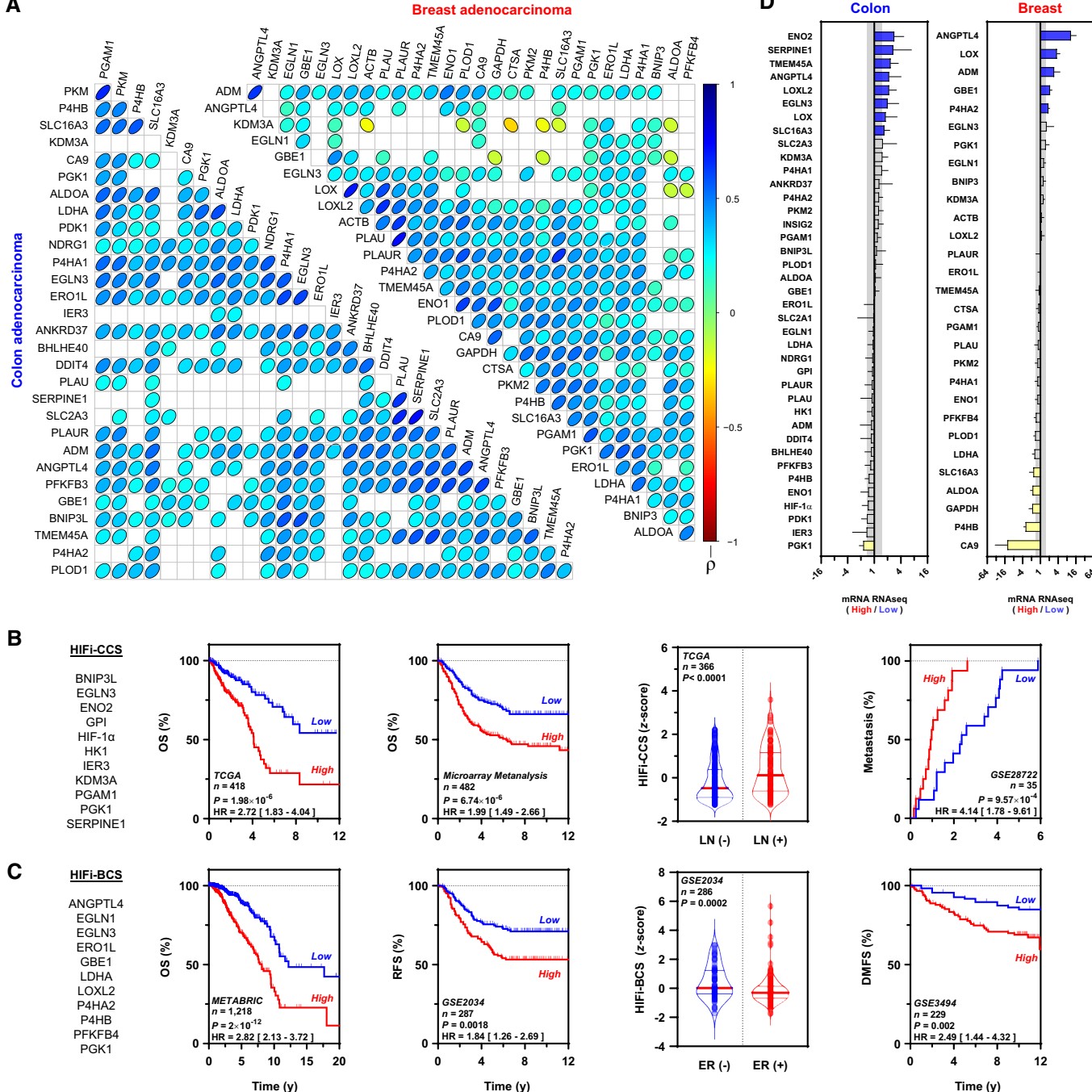

**Figure 4.**

◄

**Figure 4. HIFα transcriptional activation in human colon and breast cancers.**

A  Correlation matrix among statistically over-represented and biologically validated HIFα targets in TCGA colon or breast cancer data. *Spearman* (ρ) coefficients were encoded in a pseudo-color scale wherein flattening of ellipses denotes increasing |ρ| values; blank cells indicate non-significant correlation pairs after Bonferroni *post hoc* comparisons. Transcripts are annotated using official gene symbols.

B  HIFα-inducible colon cancer signature (HIFi-CCS). *Left and middle/left*: List of HIFα targets and overall survival in TCGA (*left*) or aggregated microarray data (*middle/left*) from colon adenocarcinomas. Cases are classified as low (blue) or high (red) HIFi-CCS according to their relationship to the median. Kaplan–Meier analysis followed by log-rank test. *Middle/right*: HIFi-CCS and lymph node (LN) invasion status. Data are standardized as z-scores. $t_{194}$ = 4.399 and $P < 0.0001$ by Student *t*-test with Welch's correction. *Right*: HIFi-CCS is associated with accelerated distant metastasis. Cases are classified as low (blue) or high (red) HIFi-CCS according to their relationship to the median. Kaplan–Meier analysis followed by log-rank test.

C  HIFα-inducible breast cancer signature (HIFi-BCS). *Left and middle/left*: List of HIFα targets and overall survival in METABRIC (*left*) or recurrence-free survival in microarray (*middle/left*) from breast adenocarcinomas. Cases are classified as low (blue) or high (red) HIFi-BCS according to their relationship to the median. Kaplan–Meier analysis followed by log-rank test. *Middle/right*: HIFi-BCS and estrogen receptor (ER) status. Data are standardized as z-scores. $t_{112}$ = 3.789 and $P = 0.0002$ by Student *t*-test with Welch's correction. *Right*: HIFi-BCS is associated with worsened distant metastasis-free survival. Cases are classified as low (blue) or high (red) HIFi-BCS whenever they are below or above the distribution median. Kaplan–Meier analysis followed by log-rank test.

D  Median RNAseq expression of individual HIFα targets as split by median HIFi-CCS (*left*) or HIFi-BCS (*right*). Transcripts up/downregulated by ± 1.5-fold are shown in blue or yellow, respectively. Error bars indicate 95% CI of the median. $Log_2$ scale; $n$ = 209 (colon, TCGA) or $n$ = 609 (breast, METABRIC).

Data information: Gene expression omnibus accession numbers are indicated; *HR*, hazard ratios, brackets indicate 95% CIs; *DMFS*, distant metastasis-free survival; *OS*, overall survival; *RFS*, recurrence-free survival; *n*, number of patients. Violin plots present 50th (thick line), 25th, and 75th percentiles (thin lines) Blue, low signature index (below median); red, high signature index (above median).

Source data are available online for this figure.

chemotherapies, similarly to colonic metastases to the liver. In addition, measurements of intra-metastatic pimonidazole$^+$, CD31$^+$ and intravascular dextran$^+$ signals revealed decreased HIF-1α levels upon LDM chemotherapy that were associated with attenuated hypoxia and HIFα-sensitive microvessel densities, independently of systemic lung perfusion. Importantly, microvessel density serves as a pathobiological correlate of both HIFα-dependent transcriptional activation via angiogenic targets, and poor clinical prognosis (Hlatky *et al*, 2002; Rey & Semenza, 2010; Schito & Rey, 2017; Schito, 2019), thus linking HIFα activity with the previously observed pre-clinical benefit of LDM chemotherapies in the EMT6/CDDP metastatic breast cancer model (Shaked *et al*, 2016).

These findings are consistent with *in vitro* studies indicating that chemotherapy can induce HIFα levels even in non-hypoxic cancer cell lines, a phenomenon thought to be correlated with cancer stem cell enrichment leading to therapy resistance, tumor recurrence, and metastasis (Samanta *et al*, 2014); likewise, a previous *in vivo* study showed that MTD doxorubicin increased HIF-1α levels in isogenic breast cancer orthografts, independently of hypoxic severity (Cao *et al*, 2013). HIF-2α induction, by contrast, has been shown to counteract HIF-1α-dependent cell cycle arrest in renal cell carcinoma lines, thereby resulting in enhanced proliferation dependent on c-Myc gain-of-function (Gordan *et al*, 2007). Importantly, our data show that LDM capecitabine dramatically offsets these effects, since the replacement of MTD capecitabine with LDM capecitabine as monotherapy or, in combination with LDM cyclophosphamide, leads to a striking decrease in HIFα levels in established (≈629 μm; HIF-1α) liver or incipient (≈180 μm; HIF-2α) lung metastases from colon and established lung metastases from breast primary tumors (≈626 μm; HIF-1α). These effects were heretofore thought to be uniquely achievable by HIFα inhibitors (i.e., topoisomerase antagonists such as topotecan, digoxin, and anthracyclines such as adriamycin) that have been occasionally administered at low dose or near-LDM regimens in combination with MTD cytotoxic agents (Rapisarda *et al*, 2004; Lee *et al*, 2009; Schito *et al*, 2012). In addition, we report that LDM administration *per se* is able to decrease HIF-1α or HIF-2α levels, wherein paralog specificity is dependent on

the interaction of factors such as tumor size, cell-autonomous, and microenvironmental effects. It is noteworthy to highlight that large, established metastases selectively upregulated HIF-1α upon MTD therapies independently of primary cell-type [i.e., HT29 (colon) versus EMT6-CDDP (breast)], secondary location (i.e., liver versus lung), and chemotherapeutic drug context (i.e., neoadjuvant versus adjuvant therapies), whereas HIF-2α was induced only in small nodules, as observed in the colonic adenocarcinoma model. These results are relevant in view of the recent development and clinical validation of HIF-2α inhibitors targeting clear cell renal cell carcinomas (Cho *et al*, 2016), and thus warrant further studies on the applicability of combinatorial LDM + paralog-specific HIFα antagonists in hypoxic and/or HIFα overexpressing cancers. The data herein presented uncover a hitherto unrecognized influence of the metastatic microenvironment on HIFα paralog expression and suggest that HIF-1α and HIF-2α do not play completely redundant roles while promoting metastatic progression in distant sites, even in oligoclonal, advanced colon and breast cancer models. In order to facilitate the translation of these targeting strategies, we developed HIFi-CCS and HIFi-BCS, two novel, biologically derived, compact HIFα transcript signatures able to predict overall survival and metastatic dissemination in colon and breast cancer patients, potentially amenable for stratification and identification of patients that are more likely to benefit from LDM alone or in combination with HIFα antagonists [e.g., HIF-2α antagonists such as PT-2385, currently under phase I trials in advanced clear cell renal cell carcinoma (Courtney *et al*, 2018)].

The translational potential of the data hereby presented can be better illustrated in light of the observation that the most compelling clinical successes with metronomic chemotherapy regimens at the pivotal randomized phase III trial level, all involve protocols wherein patients receive conventional, upfront MTD therapy followed by long-term "maintenance" regimens, not unlike continuous LDM chemotherapy. Therefore, the ability of LDM chemotherapies to offset MTD-triggered HIFα upregulation uncovers a molecular mechanism potentially subjacent to the favorable clinical profile of maintenance LDM therapies, as observed in phase III

clinical trials of advanced colorectal adenocarcinomas [ClinicalTrials.gov ID NCT00442637; (Simkens et al, 2015)] and high-risk pediatric rhabdomyosarcomas (Bisogno et al, 2018). Furthermore, there is no a priori rationale precluding favorable clinical results in other solid malignancies (i.e., triple-negative breast cancers receiving upfront MTD chemotherapies) wherein maintenance LDM therapies have been shown to result in improved clinical outcomes as well (Kerbel & Grothey, 2015; Colleoni et al, 2016; André et al, 2019).

# Materials and Methods

## Cell lines

Human colonic (HT29, ATCC) and cisplatin-resistant mouse mammary adenocarcinoma [EMT6-CDDP; (Teicher et al, 1990)] cells were grown in RPMI-1640 or high-glucose DMEM media (Gibco), respectively, supplemented with 10% FBS (HyClone) and authenticated by short-tandem repeat DNA profiling (Genetica DNA Laboratories). Cells were passaged for < 4 months after being authenticated and routinely tested negative for *Mycoplasma spp*.

## Models of advanced metastatic cancer

Orthotopic colon adenocarcinomas were established by implantation of subcutaneous HT29 tumors expressing the *p*GL3 firefly luciferase vector (Promega), excised, and dissected into 3–5 mm$^3$ pieces and orthotopically sutured onto the abluminal *caecal* wall of 6- to 8-week-old male immunodeficient SCID mice (Hackl et al, 2013; Shaked et al, 2016). Tumor growth was quantified every 7 days as *in vivo* bioluminescence (Xenogen, IVIS imaging). Mice were euthanized after 7 weeks of chemotherapy (luminescence > 10$^7$ photons/s); thereafter, colon, lungs, and livers were resected *in toto*, fixed, and embedded for quantitative immunohistochemistry. Syngeneic metastatic breast cancers were established by implanting EMT6-CDDP cells (10$^5$) into the inguinal mammary fat pad of 6- to 8-wk-old female immunocompetent BALB/J mice (Jackson Labs) (Teicher et al, 1990; Shaked et al, 2016). Primary breast tumors were allowed to grow for 12 days (median volume ≈270 mm$^3$, 95% CI: 256–285; $n$ = 50) and resected before initiating adjuvant LDM and MTD chemotherapy. Adjuvant treatment was maintained for 10 days while monitoring for signs of overt metastatic disease (*e.g.,* labored breathing, ascites, ulceration of residual primary tumors, hindlimb paralysis, or ≥ 20% weight loss). To assess tissue perfusion, mice were injected with fluorescein-labeled dextran (100 mg/kg IV; FD-150S, MW 150 kD, Millipore-Sigma) dissolved in 0.9% saline solution, within 5–15 min before euthanization. All surgical procedures were undertaken in accordance with the animal care guidelines of Sunnybrook Health Sciences Centre and the Canadian Council of Animal Care.

## Chemotherapeutic drug treatments

Experimental chemotherapy regimens were started 3 weeks (HT29) or 12 days (EMT6-CDDP) after tumor implantation. Cyclophosphamide (Baxter) was administered at 20 mg/kg/day PO through the drinking water (Man et al, 2002), whereas capecitabine (LC Laboratories) was prepared in a solution containing 20 mg/ml

hydroxypropyl cellulose (Klucel-LF, Ashland), 0.9 mg/ml Methyl-P, 0.1 mg/ml Propyl-P, and 0.1% Tween-80 (Sigma-Aldrich) at a LDM dose of 100 mg/kg/day PO by gavage or at an MTD dose of 400 mg/kg/day PO for 4 days, followed by a 17 days drug-free break period, when appropriate.

## Quantitative immunohistochemistry

Tissue processing and antigen retrieval were performed as previously described (Schito et al, 2012); briefly, 5-μm thick sections were incubated with primary anti-HIF-1α, HIF-2α, CA9, pimonidazole, CD31, or Ki67 antibodies (Appendix Table S1), further processed with a DAB-based protocol (VECTASTAIN Elite ABC-HRP Kit; Vector Laboratories) and counterstained with *Mayer*'s hematoxylin (Sigma). Negative controls were implemented by replacing primary antibodies with isotype-matched IgGs. Liver and kidney sections served as positive controls for HIF-1α or HIF-2α. Microphotographs (≈9,300 digital images) were acquired at resolutions of 5.2 (×2.5), 0.7 (×20), or 0.3 (×40) μm/pixel and stitched against a white digital canvas. Unsupervised, automated quantitative immunohistochemistry (qIHC) was performed with custom macros coded in ImageJ (v1.52p, NIH), with the exception of Ki67-based proliferation indexes, herein measured via a machine-learning algorithm (Schüffler et al, 2013). Manual delineation of individual metastatic regions was followed up by automated isolation into individual images, submitted to morphometric and staining quantification. HIF-1α, HIF-2α, CA9, pimonidazole, and CD31 were measured as fractions of primary tumoral or metastatic cross-sectional area at fixed pixel intensity thresholds. Ki67 proliferative indexes were calculated as the fraction of positive *nuclei* among 18−52 random high-power (×20) fields/tumor. The correlation between tumoral immunoreactivities and their colocalization was determined in consecutive tissue sections after automatic alignment. To assess perfusion of lung tissue within the metastatic breast cancer model, *nuclei* were labeled with 1 μg/ml Hoechst-33342 in PBS pH 7.8 for 10 min at room temperature and imaged under DAPI and FITC filters to determine lung perfused areas as a fraction of total lung tissue cross-sectional area; fluorescence images were digitally stitched against a square black canvas.

## HIFα transcriptomic signature and patient survival analysis

A biologically derived HIFα signature was developed on the basis of experimentally validated transcripts from a previously published microarray colon cancer hypoxia signature [CCHS (Dekervel et al, 2014)]. We noticed that, in addition to HIFα-independent, hypoxia-inducible transcripts, CCHS contained five canonical "seed" HIFα targets (i.e., BNIP3, DDIT4, P4HA1, P4HA2, and PLAUR). Consequently, we analyzed RNAseq data within the TCGA colonic adenocarcinoma repository by reasoning that CCHS "seed" genes would randomly correlate with < 7 HIFα targets among the top 200 ranked transcripts (by *Spearman* ρ), since HIFα genes represent ≈2% of the transcriptome (Manalo et al, 2005), thereby yielding $P < 0.049$ for ≥7 hits, assuming a binomial statistical distribution. In order to generate a compact HIFα signature with maximal predictive power, significant HIFα targets ($P < 0.01$ after Bonferroni corrections) were submitted to

stepwise regression followed by a *Cox* proportional hazards model using overall survival as the dependent variable, subjected to log-rank tests, implemented in *R* [v3.6.2] (Venables & Ripley, 2002). Furthermore, we cross-validated TCGA-derived signatures using colon and breast cancer microarray metadata [*SurvExpress* (Aguirre-Gamboa *et al*, 2013)] and data obtained from the NCBI gene expression omnibus (GEO). The following publicly available colon and breast cancer datasets were used: TCGA colonic adeno-carcinoma (COAD; Muzny *et al*, 2012b; Muzny *et al*, 2012a), GSE28722 (Loboda *et al*, 2011b; Data ref: Loboda *et al*, 2011a), METABRIC (Curtis *et al*, 2012b; Data ref: Curtis *et al*, 2012a), GSE2034 (Wang *et al*, 2005b; Data ref: Wang *et al*, 2005a), and GSE3494 (Miller *et al*, 2005b; Data ref: Miller *et al*, 2005a).

### Statistical analysis

All data are expressed as medians, quartiles, and 95% confidence intervals, while *n* indicates the number of biological replicates or metastatic nodules in each experiment. For liver and lung metastatic nodule analysis, *n* indicates the number of nodules, and the number of mice per experimental group is also provided. Logarithmic trans-formation of fractional data was used to ensure Gaussian distributions prior to statistical analyses. Survival curves were depicted as Kaplan–Meier analyses followed by log-rank tests. Differences between two experimental groups were assessed with Student *t*-tests with Welch's corrections, while three or more groups were evaluated by one-way Brown–Forsythe ANOVA followed by Bonferroni or Benjamini, Krieger and Yekutieli *post hoc* comparisons ($\alpha = 0.05$). Categorical tumor size data were analyzed with two-sided chi-square tests. Differences among nonlinear and linear model fit parameters were assessed with *F*-tests; *Pearson r* and *Spearman* $\rho$ coefficients are shown for linear regression and transcriptomic correlations, as appropriate.

## Data availability

This study includes no data deposited in external repositories.

**Expanded View** for this article is available online.

## Acknowledgements
We would like to thank Petia Stefanova (histology core facility at SRI) for superb histological and technical assistance. The results published herein are in part based on data generated by the TCGA Research Network: https://www.cancer.gov/tcga. This study was supported by grants from the Canadian Institutes of Health Research (CIHR) and the Canadian Breast Cancer Foundation (to RSK). LS and SR are Fellows of the UCD Conway Institute of Biomolecular & Biomedical Research, supported by the UCD *Ad Astra* Fellows Programme and the UCD School of Medicine, University College Dublin, Ireland.

## Author contributions
LS performed qIHC. LS and SR analyzed data, and performed statistical modeling and transcriptomic analyses. SR developed coding tools. PX and SM performed animal experiments, resected, processed, and stored tissues for qIHC. WC-M performed intravenous fluorescently labeled dextran injections for perfusion experiments. LS and SR wrote the manuscript. LS, SR, and RSK designed experiments and reviewed the final manuscript with input from all authors. All authors read and approved the final manuscript.

## Conflict of interest
The authors declare that they have no conflict of interest.

---

**The paper explained**

**Problem**

Hypoxia, a common feature of most solid cancers, arising as a mismatch between cellular oxygen demand and supply, is associated with unfavorable chemotherapeutic responses, recurrence, and cancer cell dissemination (metastasis). Conventional chemotherapy, a systemic treatment for many cancers, relies upon the administration of therapeutics at maximum tolerated dose (MTD) that can massively eliminate cancer cells over a relatively short period of time. Nevertheless, this treatment modality can further increase hypoxia, while creating microenvironmental conditions that enable cancer cell regrowth and dissemination, in addition to causing toxic side effects. By contrast, low-dose, frequent, and regular administration of chemotherapy drugs often referred to as metronomic chemotherapy, causes fewer adverse effects, while displaying therapeutic benefits in a variety of preclinical models of cancer.

**Results**

In this paper, we performed the first quantitative side-by-side comparison of metronomic and conventional chemotherapies upon tumoral hypoxia and hypoxia-inducible factors (HIFs), a family of master transcriptional regulators exerting pleiotropic functions on many aspects of cancer biology, including therapy resistance, metastasis, and mortality. We found that metronomic chemotherapy dramatically improves tumor oxygenation, while offsetting HIF induction caused by conventional chemotherapy regimens in preclinical models of metastatic colon and breast cancers. In addition, we developed two compact, 11-gene signatures that allow a simplified estimation of HIF levels in colon and breast cancers, which might be helpful to identify patients more likely to benefit from metronomic therapies.

**Impact**

These results provide a rationale for the use of maintenance metronomic chemotherapy strategies to improve oxygenation and decrease cancer dissemination in two of the most common types of cancer, while adding another tool to the armamentarium aimed at improving survival in patients bearing hypoxic cancers, possibly by combining metronomic chemotherapies with novel compounds capable of directly targeting HIFs.

---

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
