## [Review Process File · EMBO Molecular Medicine]

Metronomic chemotherapy offsets HIF α induction upon maximum-tolerated dose in metastatic cancers

Luana Schito, Sergio Rey, Ping Xu, Shan Man, William Cruz-Muñoz and Robert S. Kerbel
DOI: 10.15252/emmm.201911416

Corresponding authors: Luana Schito (luana.schito@ucd.ie) , Robert Kerbel (robert.kerbel@sri.utoronto.ca)

Review Timeline:

Submission Date:	7th Sep 19
Editorial Decision:	8th Oct 19
Revision Received:	12th May 20
Editorial Decision:	2nd Jun 20
Revision Received:	21st Jun 20
Accepted:	23rd Jun 20

Editor: Lise Roth

Transaction Report:

8th Oct 2019

Dear Dr. Schito,

Thank you for the submission of your manuscript to EMBO Molecular Medicine. We have now received feedback from the three reviewers who agreed to evaluate your manuscript. As you will see from the reports below, the referees acknowledge the interest of the study and are overall supporting publication of your work pending appropriate revisions.

Addressing the reviewers' concerns in full will be necessary for further considering the manuscript in our journal, and acceptance of the manuscript will entail a second round of review. We realize that this represents a lot of additional work, but given the interest of the findings, we are willing to extend the revision timeframe to 6 months.

EMBO Molecular Medicine encourages a single round of revision only and therefore, acceptance or rejection of the manuscript will depend on the completeness of your responses included in the next, final version of the manuscript. For this reason, and to save you from any frustrations in the end, I would strongly advise against returning an incomplete revision.

When submitting your revised manuscript, please carefully review the instructions that follow below. Failure to include requested items will delay the evaluation of your revision:

2) Individual production quality figure files as .eps, .tif, .jpg (one file per figure).

3) A .docx formatted letter INCLUDING the reviewers' reports and your detailed point-by-point responses to their comments. As part of the EMBO Press transparent editorial process, the point-by-point response is part of the Review Process File (RPF), which will be published alongside your paper.

4) A complete author checklist, which you can download from our author guidelines (<https://www.embopress.org/page/journal/17574684/authorguide#submissionofrevisions>). Please insert information in the checklist that is also reflected in the manuscript. The completed author checklist will also be part of the RPF.

6) Before submitting your revision, primary datasets produced in this study need to be deposited in an appropriate public database (see <https://www.embopress.org/page/journal/17574684/authorguide#dataavailability>). Please remember to provide a reviewer password if the datasets are not yet public. The accession numbers and database should be listed in a formal "Data Availability" section

(placed after Materials & Method). Please note that the Data Availability Section is restricted to new primary data that are part of this study.

7) We would also encourage you to include the source data for figure panels that show essential data. Numerical data should be provided as individual .xls or .csv files (including a tab describing the data). For blots or microscopy, uncropped images should be submitted (using a zip archive if multiple images need to be supplied for one panel). Additional information on source data and instruction on how to label the files are available at .

8) Our journal encourages inclusion of *data citations in the reference list* to directly cite datasets that were re-used and obtained from public databases. Data citations in the article text are distinct from normal bibliographical citations and should directly link to the database records from which the data can be accessed. In the main text, data citations are formatted as follows: "Data ref: Smith et al, 2001" or "Data ref: NCBI Sequence Read Archive PRJNA342805, 2017". In the Reference list, data citations must be labeled with "[DATASET]". A data reference must provide the database name, accession number/identifiers and a resolvable link to the landing page from which the data can be accessed at the end of the reference. Further instructions are available at .

9) We replaced Supplementary Information with Expanded View (EV) Figures and Tables that are collapsible/expandable online. A maximum of 5 EV Figures can be typeset. EV Figures should be cited as 'Figure EV1, Figure EV2" etc... in the text and their respective legends should be included in the main text after the legends of regular figures.

- Additional Tables/Datasets should be labeled and referred to as Table EV1, Dataset EV1, etc. Legends have to be provided in a separate tab in case of .xls files. Alternatively, the legend can be supplied as a separate text file (README) and zipped together with the Table/Dataset file. See detailed instructions here: .

10) The paper explained: EMBO Molecular Medicine articles are accompanied by a summary of the articles to emphasize the major findings in the paper and their medical implications for the non-specialist reader. Please provide a draft summary of your article highlighting

11) For more information: There is space at the end of each article to list relevant web links for further consultation by our readers. Could you identify some relevant ones and provide such information as well? Some examples are patient associations, relevant databases,

OMIM/proteins/genes links, author's websites, etc...

12) Every published paper now includes a 'Synopsis' to further enhance discoverability. Synopses are displayed on the journal webpage and are freely accessible to all readers. They include a short stand first (maximum of 300 characters, including space) as well as 2-5 one-sentence bullet points that summarize the paper. Please write the bullet points to summarize the key NEW findings. They should be designed to be complementary to the abstract - i.e. not repeat the same text. We encourage inclusion of key acronyms and quantitative information (maximum of 30 words / bullet point). Please use the passive voice. Please attach these in a separate file or send them by email, we will incorporate them accordingly.

Please also suggest a striking image or visual abstract to illustrate your article. If you do please provide a jpeg file 550 px-wide x 400-px high.

13) As part of the EMBO Publications transparent editorial process initiative (see our Editorial at <http://embomolmed.embopress.org/content/2/9/329>), EMBO Molecular Medicine will publish online a Review Process File (RPF) to accompany accepted manuscripts.

In the event of acceptance, this file will be published in conjunction with your paper and will include the anonymous referee reports, your point-by-point response and all pertinent correspondence relating to the manuscript. Let us know whether you agree with the publication of the RPF and as here, if you want to remove or not any figures from it prior to publication.

EMBO Molecular Medicine has a "scooping protection" policy, whereby similar findings that are published by others during review or revision are not a criterion for rejection. Should you decide to submit a revised version, I do ask that you get in touch after six months if you have not completed it, to update us on the status.

I look forward to receiving your revised manuscript.

Yours sincerely,

Lise Roth

Lise Roth, PhD
Editor
EMBO Molecular Medicine

To submit your manuscript, please follow this link:

Link Not Available

***** Reviewer's comments *****

Referee #1 (Remarks for Author):

In the manuscript "Low-dose metronomic chemotherapy offsets HIF α upregulation in metastatic colon cancer" the authors show that low-dose metronomic (LDM) capecitabine is able to offset HIF α upregulation in advanced colorectal adenocarcinomas and their liver metastases, and HIF2 α upregulation in lung metastases, caused by conventional maximum tolerated dose (MTD) chemotherapy. The authors suggest that the maintenance of LDM therapies can potentially improve prognosis and clinical outcomes in patients with advanced colorectal adenocarcinomas. Overall, the manuscript is well written and presents data with translational potential, however the advantages of metronomic schemes in vivo are poorly characterized and should be stressed. Several conceptual issues somehow weaken the message conveyed by the authors, and those should be addressed.

Major concerns

LDM cyclophosphamide + LDM capecitabine compared to LDM cyclophosphamide + MTD capecitabine regimen induces a significant decrease of HIF1 α but not in metastatic diameter. In this context the clinical advantages of LDM scheme are not so clear. There is not a straightforward correlation between HIF1 α and metastatic size. How do the authors explain these differences? The authors might evaluate the liver metastatic diameter and HIF1 α expression at middle stage.

In Fig 3B, the authors show that only LDM capecitabine decreases nodule size whereas all other treatments either increase or leave lung metastatic size unaffected. Furthermore, in LDMCPB regimen compared to vehicle, the reduction of metastasis size is inversely correlated with the increase in HIF2 α . How do the authors explain these effects? The authors should clarify why compared to vehicle LDM cyclophosphamide + LDM capecitabine regimen does not reduce lung metastatic diameter. The metronomic scheme for the combination cyclophosphamide/capecitabine does not look like so clinical favorable.

So far, the authors analyze the effects of LDM and MTD chemotherapies on metastasis size. Which are the effects on metastatic count? Please define and include in all figures the representative micrographs.

To support the clinical relevance of metronomic chemotherapy, the in vivo experiments should be performed also in a syngeneic mouse model of colorectal adenocarcinoma.

Furthermore, the authors should evaluate the effects of LDM and MTD chemotherapies on the overall survival of mice. Does the low-dose metronomic chemotherapy improve the survival of mice?

To strengthen further the advantages of metronomic therapies on HIF1 α downregulation, the authors should evaluate the vessel perfusion by lectin-FITC and the tumor hypoxic area by pimonidazole and by MRI-Oxy enhanced imaging.

Minor concerns

In "Materials and Methods" paragraph, the authors should explain how they inspect and quantify metastasis size.

The authors should include in Fig 1G the y axis title.

The authors should include in Fig 3C the legend of colors.

In all figures the representative micrographs are incomplete, LDMCPB, MTDCPB, LDMCTX treatments should be included.

Treatment schedules behind the data should be included in the figures.

Referee #2 (Remarks for Author):

The manuscript investigates and compares the impact of maximum tolerated dose chemotherapy and low-dose metronomic chemotherapy on tumour hypoxia in particular on the stabilisation of HIF1 α and HIF2 α .

The study shows increased HIF1 α (in primary tumours and liver metastasis) and HIF2 α (in lung metastases) in response to capecitabine maximum tolerated dose therapy that was not seen in the low-dose metronomic chemotherapy regimen.

Further to this the clinical significance of HIF1 α and HIF2 α transcriptionally regulated targets was investigated in RNA seq and RNA array data sets including the development of an 11 gene signature.

The manuscript is well written and aspects of this research add significantly to the understanding of the impact of maximum tolerated dose therapy on hypoxic signalling.

Comments:

1. In Figure 1a there is a clear increase in HIF1 α stabilisation in response to capecitabine maximum tolerated dose treatment however in Figure 1c no increase in the well-described and documented HIF1 α transcriptional target is seen.

This suggests that there may be no functional impact (e.g. transcriptional response) of HIF1 α stabilisation in this setting, a fact that requires either further investigation of other HIF1 α transcriptional targets which clarify whether the additional HIF stabilisation is of functional relevance or a clearer explanation of this discrepancy.

The response in the text " these data suggest the presence of cell-autonomous effects, highlighting the need for caution whenever CA9 is used as a substitute for HIF α levels in hypoxic cancers." is insufficient to explain this lack of effect.

2. It is unclear how the 38 gene and/or the 11 gene signature were identified. These were proposed to be derived from the Dekerveil et al 2014 manuscript however this identifies a 21 gene signature? The value of what these new 38 gene and/or the 11 gene signatures add to literature (given the other published hypoxia gene signatures) also requires additional support in the text. This should include more detail on how these signatures could help identify patients likely to benefit from LDM alone or in combination with HIF1 α inhibition, as no detailed explanation of this is given.

Referee #3 (Remarks for Author):

The manuscript "Low-dose metronomic chemotherapy offsets HIF1a upregulation in metastatic colon cancer" by Schito et al. aims to define the differential impact of maximum tolerated dose (MTD) and low-dose metronomic (LDM) chemotherapy on spontaneous metastases arising from orthotopically implanted colon cancer fragments. The authors report organ-specific functions of HIFa paralogs, where treatment with MTD capecitabine induced the expression of HIF-1a in primary tumors and liver metastasis and HIF-2a in lung metastasis. Indeed, treatment with LDM capecitabine could reverse the observed induction of organ-specific expression of HIFa paralogs. Further, by reanalyzing published datasets, the authors defined an 11-gene predictive signature to stratify colon cancer patients for LDM or HIF-1 α -targeting therapies.

Employing a state-of-the-art mouse metastasis model, the manuscript reveals paralog-specific functions of HIFa during metastasis. This report presents the first side-by-side comparison of MTD and LDM therapy for metastatic colon cancer. The manuscript is conceptually at the cutting edge of translational cancer biology. It is very timely for the mechanistic understanding of the underlying differences between MTD and LDM therapies. Yet, as presented, the study falls short of supporting all of the authors' claims and it is felt some more definite experiments will be required to substantiate some of the findings. Pending satisfactory revision, this could become a very substantial and important contribution to the journal.

In further advancing the work, the authors are particularly encouraged to consider the following:

1. The findings of this manuscript present to the best of the reviewer's knowledge the first side-by-side preclinical comparison of MTD and LDM therapy on metastatic colon cancer. Even though the reviewer appreciates the use of a spontaneously-metastasizing colon cancer model in this study, it will be crucial to validate the findings in a second tumor model.
2. Different chemotherapy regimens did not affect the growth of primary tumors (Suppl Fig. 1A). Nevertheless, the Ki67-dependent proliferation index shows distinct patterns of tumor cell proliferation for different therapeutic regimens (Fig. 1G), especially with the vehicle and the combination of LDM-CTX and MTD-CDB exhibiting a similar score. This dichotomy between different measurement techniques should be addressed and discussed appropriately. Additionally, the authors should consider performing the analysis of the Ki67-proliferation index at an earlier time point (around 2 weeks post-therapy initiation, which will be roughly equivalent to one doubling time of tumors).
3. Concerning Fig 2E, how did the authors distinguish between parenchymatous and metastatic tissue? Could it be that there are some locally-invading colon cancer cells in the parenchymal tissue? To strengthen the conclusions, the author should consider co-staining HIF-1a and HIF-2a with a cancer cell- (or hepatocyte-) specific marker to unambiguously mark the metastatic area.
4. The median diameter of a metastatic nodule in the liver was 689 μ m, whereas the lung was 211 μ m. The authors mention on page 7 that the differential induction of HIFa-paralogs might suggest their specific roles during different stages of metastatic colonization. To validate this hypothesis, it would be crucial to judge this in (possibly) an experimental model in which lung metastatic nodules can be allowed to reach approx. 700 μ m. This might further add a crucial explanation to the observation that the lung metastatic nodule size was reduced in the LDM-CPB treatment group but was found increased in the combination (LDM-CPB + LDM-CTX) treatment

group (referring to Fig.3C).

5. Stratification of metastases-bearing patients is a clinically unmet challenge and has historically been one of the major bottlenecks for testing stroma-targeting therapies. By carefully reanalyzing previously-published datasets, the authors have defined an 11-gene signature to predict the vulnerability of tumor cells towards LDM chemo- or HIFa-targeting therapies. To further strengthen their findings, the authors should consider evaluating the expression of the established panel of genes in their preclinical spontaneous metastasis model, preferably comparing the primary tumor with the multiorgan metastases.

6. The reviewer appreciates the stringent analyses performed by the authors to investigate metastatic nodular diameter (stratification based on the median of the control group) and the expression of HIFa-paralogs following different therapeutic regimens (Fig. 2 A-C and 3 A-C). The authors should describe the method in the main text, thereby setting a benchmark for future image analysis in similar mouse metastatic models.

7. The authors need to carefully inspect the manuscript for editorial errors, especially in the legend of Fig. 3 where they describe the liver instead of lungs and on page 5, where they are describing Fig. 2A and mention LDM capecitabine instead of MTD capecitabine.

8. The authors need to elaborate figure legends to allow standalone interpretation of the figures.

Referee #1: In the manuscript "Low-dose metronomic chemotherapy offsets HIF α upregulation in metastatic colon cancer" the authors show that low-dose metronomic (LDM) capecitabine is able to offset HIF α upregulation in advanced colorectal adenocarcinomas and their liver metastases, and HIF2 α upregulation in lung metastases, caused by conventional maximum tolerated dose (MTD) chemotherapy. The authors suggest that the maintenance of LDM therapies can potentially improve prognosis and clinical outcomes in patients with advanced colorectal adenocarcinomas. Overall, the manuscript is well written and presents data with translational potential, however the advantages of metronomic schemes *in vivo* are poorly characterized and should be stressed. Several conceptual issues somehow weaken the message conveyed by the authors, and those should be addressed.

Major concerns

Referee #1: "LDM cyclophosphamide + LDM capecitabine compared to LDM cyclophosphamide + MTD capecitabine regimen induces a significant decrease of HIF1 α but not in metastatic diameter. In this context the clinical advantages of LDM scheme are not so clear. There is not a straightforward correlation between HIF1 α and metastatic size. How do the authors explain these differences? The authors might evaluate the liver metastatic diameter and HIF1 α expression at middle stage."

Answer: The reviewer is correct. The correlation between metastatic diameter and HIF α levels is not straightforward or direct, but rather a multi-factorial one. In order to better understand this relationship and further inquire into the reviewers' concern, we have revised the manuscript by including a completely new dataset, utilizing an immunocompetent, orthotopic breast cancer model that readily metastasizes to the lungs after primary resection. It is worth to mention that adjuvant LDM chemotherapy with capecitabine + cyclophosphamide has been demonstrated to significantly increase survival in this model (Shaked et al. – PMID: 27569209). Analysis of metastatic lung nodules that were collected at practically identical metastatic diameters (within $\approx 1\%$ difference by automated morphometry) showed increased HIF-1 α levels under MTD regimens that were offset by LDM in both models. Importantly, this extensive set of new experiments further supports the notion that metastatic diameter *per se* is not a strong predictor of mortality in the clinical setting (e.g., an extreme example are metastatic seminomas, amenable for treatment and cures with radiotherapy albeit in the presence of large brain metastases). Moreover, our work is consistent with previous studies showing that HIF-1 α induction can induce cell cycle arrest in HCT116 colon adenocarcinoma cells *in vitro* (Hubbi et al. – PMID: 23405012); indeed, the current study is the first *in situ* instance wherein HIF-1 α expression has been found to be inversely correlated with Ki67 in primary orthotopic colonic tumors, further confirmed by the second advanced breast cancer model, as presented in this revision.

Importantly, this revised manuscript illustrates that there is indeed an overall effect of metastatic diameter on HIF α^+ areas measured as μm^2 ; an effect that nonetheless disappears when normalizing by individual metastatic diameter. Moreover, independence of size was further confirmed by measurements of intra-metastatic tumor hypoxia, strongly suggesting that microenvironmental and cell-autonomous mechanisms acting on HIF α are modulated by chemotherapeutic modalities, independently of tumor diameter. Consistently, our results suggest that tumor size is not sufficient to explain HIF α levels in primary tumors and their metastasis, since LDM chemotherapy significantly decreased HIF α levels independently of size in both preclinical models. These results are in agreement with a large body of data indicating that HIF α , rather than metastatic or primary tumor size, acts as an independent predictor of mortality in cancer patients (reviewed by Schito and Semenza – PMID: 28741521).

Referee #1: In Fig 3B, the authors show that only LDM capecitabine decreases nodule size whereas all

other treatments either increase or leave lung metastatic size unaffected. Furthermore, in LDMCPB regimen compared to vehicle, the reduction of metastasis size is inversely correlated with the increase in HIF2 α . How do the authors explain these effects? The authors should clarify why compared to vehicle LDM cyclophosphamide + LDM capecitabine regimen does not reduce lung metastatic diameter. The metronomic scheme for the combination cyclophosphamide/capecitabine does not look like so clinical favorable.

Answer: In part, this question is connected to the previous answer. It is important to note that our observations are not unlike common clinical findings in the oncological setting, wherein chemotherapies are able to stall growth in one metastatic location whilst being ineffective in another. Importantly, these observations can be understood in light of *in vitro* data indicating that HIF α paralogs can have opposite effects on cellular proliferation (Gordan *et al.* – PMID: 17418410). That is, HIF-1 α induces cell cycle arrest, whilst HIF-2 α promotes proliferation. Paralog-specific effects in particular metastatic sites, do highlight the interaction between the metastatic niche microenvironment and therapeutic regimens/modalities, resulting in HIF-1 α or -2 α induction. It is worth to consider that lung metastases from HT29 tumors were significantly smaller than liver nodules. Notwithstanding, we have in the interim improved the performance of our quantification algorithm with the ultimate goal of making it completely automated. Quantification of metastatic size and dichotomization according to the vehicle median diameter show that both capecitabine and cyclophosphamide monotherapies significantly decrease nodule size in lung metastases from the HT29 colon adenocarcinoma model (Fig. EV1A). Furthermore, lung metastases from the new EMT6-CDDP breast model behaved similarly to liver HT29 metastases, in addition to being almost identical in size; in particular, LDM monotherapies and the doublet LDM regimen were able to significantly decrease HIF-1 α levels (Fig. 3C). Interestingly, lung metastases in the breast model did not express HIF-2 α . Taken together, these results suggest that HIF-1 α and -2 α exert non-redundant effects on metastasis that are dependent on the tumor microenvironment; in addition, comparison of metastasis with similar size from two different primary tumors and preclinical models (HT29 versus EMT6-CDDP) suggests that HIF-1 α induction occurs preferentially in larger metastasis.

Referee #1: So far, the authors analyze the effects of LDM and MTD chemotherapies on metastasis size. Which are the effects on metastatic count? Please define and include in all figures the representative micrographs.

Answer: In the revised manuscript, we have included nodule counts within all dichotomized nodule diameter graphs. Please see Fig. 2A, Fig. EV1A and Fig. 3A. In general, we observe a decrease in metastatic counts with chemotherapies and, in some cases, an increase in MTD capecitabine treated groups; unfortunately, statistical testing of nodule numbers would require a much larger study, that would be ethically unjustifiable as the sole purpose of an additional experimental series. As suggested by Referee #1, we have included micrographs for all groups generated from the previous as well new experiments. Please see: Fig. 3B-C; Appendix Fig. S1A-B; Appendix Fig. S4A-B; Appendix Fig. S5A, and Appendix Fig. S6A.

Referee #1: To support the clinical relevance of metronomic chemotherapy, the *in vivo* experiments should be performed also in a syngeneic mouse model of colorectal adenocarcinoma.

Answer: We performed a completely new series of experiments in a syngeneic orthotopic breast carcinoma model, which spontaneously disseminates to the lungs (EMT6-CDDP), currently in use in our lab. Lung nodules in this second model behave similarly to liver metastases from the HT29 model (see Fig. 3, Fig. EV2). These new experiments also address one of Referee #3' suggestions.

Referee #1: To strengthen further the advantages of metronomic therapies on HIF1 α downregulation, the authors should evaluate the vessel perfusion by lectin-FITC and the tumor hypoxic area by pimonidazole

and by MRI-Oxy enhanced imaging.

Answer: In this revised manuscript, we have performed lung perfusion experiments by intravenous administration of high MW fluorescein-labeled dextran in the EMT6-CDDP breast metastatic model. Importantly, results of these experiments indicate that HIF-1 α offsetting by LDM chemotherapies are independent of systemic circulatory effects. Moreover, we did not find any evidence of perfusion differences within metastases (Appendix Fig. S6A-B). Nonetheless, measurements of hypoxic fractions with pimonidazole in the EMT6-CDDP breast model indicated a significant decrease of intra-metastatic tumor hypoxia (Fig. 3B), that is independent of tissue perfusion and nodular diameters.

Minor concerns

Referee #1: In "Materials and Methods" paragraph, the authors should explain how they inspect and quantify metastasis size.

Answer: We have revised this section to further include details of the image analyses methods in use. Our intention is to write-up a full methods article in the near future, whilst automating as many image processing steps as possible. Please see "Materials and Methods" new section.

Referee #1: The authors should include in Fig 1G the y axis title.

Answer: We have added this information in the revised version of now Fig 1D.

Referee #1: The authors should include in Fig 3C the legend of colors.

Answer: We have added this information in the revised figure legend.

Referee #1: In all figures the representative micrographs are incomplete, LDM^{CPB}, MTD^{CPB}, LDM^{CTX} treatments should be included.

Answer: This has been done in order to emphasize the relevant differences among groups, since the groups that the Referee refers to present signal levels comparable or below vehicle, which in practice is difficult to appreciate when microphotographs need to be reduced to accommodate extra control groups. Nonetheless, we have included micrographs for all groups generated from the previous as well new experiments. Please see: Fig. 3B-C; Appendix Fig. S1A-B; Appendix Fig. S4A-B; Appendix Fig. S5A, and Appendix Fig. S6A.

Referee #1: Treatment schedules behind the data should be included in the figures.

Answer: We have added this information in the revised version.

Referee #1: Furthermore, the authors should evaluate the effects of LDM and MTD chemotherapies on the overall survival of mice. Does the low-dose metronomic chemotherapy improve the survival of mice?

Answer: This model of colorectal adenocarcinoma, due to its advanced metastatic nature, results in accelerated mortality, thus making it difficult to observe OS benefits; notwithstanding, doublet LDM capecitabine + cyclophosphamide has been shown to significantly improve survival in the EMT6-CDDP model (Shaked et al. – PMID: 27569209), included in this revised version.

Referee #2:

Referee #2 (Remarks for Author):

The manuscript investigates and compares the impact of maximum tolerated dose chemotherapy and low-dose metronomic chemotherapy on tumour hypoxia in particular on the stabilisation of HIF1 α and HIF2 α . The study shows increased HIF1 α (in primary tumours and liver metastasis) and HIF2 α (in lung metastases) in response to capecitabine maximum tolerated dose therapy that was not seen in the low-dose metronomic chemotherapy regimen.

Further to this the clinical significance of HIF1 α and HIF2 α transcriptionally regulated targets was investigated in RNA seq and RNA array data sets including the development of an 11 gene signature.

The manuscript is well written and aspects of this research add significantly to the understanding of the impact of maximum tolerated dose therapy on hypoxic signalling.

Answer: Thank you.

Comments:

Referee #2: 1. In Figure 1a there is a clear increase in HIF1 α stabilisation in response to capecitabine maximum tolerated dose treatment however in Figure 1c no increase in the well-described and documented HIF1 α transcriptional target is seen.

This suggests that there may be no functional impact (e.g. transcriptional response) of HIF1 α stabilisation in this setting, a fact that requires either further investigation of other HIF1 α transcriptional targets which clarify whether the additional HIF stabilisation is of functional relevance or a clearer explanation of this discrepancy.

The response in the text " these data suggest the presence of cell-autonomous effects, highlighting the need for caution whenever CA9 is used as a substitute for HIF α levels in hypoxic cancers." is insufficient to explain this lack of effect.

Answer: To the best of our knowledge, there is no published instance of HIF α upregulation resulting in across-the-board lack of hypoxic transcriptome upregulation, with the exception of the HIF-3 α dominant-negative isoform IPAS, that is expressed in mice, having a very restricted tissue expression pattern. Moreover, in line with our results, another report found lack of CA9 induction under radiochemotherapy and/or increased CA9 levels in advanced tumors (typically hypoxic) in rectal adenocarcinoma patients (Mayer et al. – PMID: 26782212) and multiple papers indicating a similar disassociation among hypoxia, HIF α and CA9 (reviewed in Pastorekova and Gillies, PMID: 31076951). We believe that it is important to report this finding, since CA9, even though one of the most consistent hypoxia inducible transcripts *in vitro*, might exhibit *in vivo* shortfalls as a *bona fide* hypoxia marker; case in point: our advanced metastatic adenocarcinoma model. In this revised manuscript version, we have introduced a second model of metastatic breast cancer and quantified microvascular densities as they correlate to HIF-1 α and intra-metastatic hypoxia. Since microvascular density integrates the transactivation of dozens of HIF α targets (Rey and Semenza, PMID: 20164116), we believe that this should suffice as an integrated response downstream of HIF α transactivity.

Referee #2: 2. It is unclear how the 38 gene and/or the 11 gene signature were identified. These were proposed to be derived from the Dekervel et al 2014 manuscript however this identifies a 21 gene signature? The value of what these new 38 gene and/or the 11 gene signatures add to literature (given the other published hypoxia gene signatures) also requires additional support in the text. This should include more

detail on how these signatures could help identify patients likely to benefit from LDM alone or in combination with HIF1 α inhibition, as no detailed explanation of this is given.

Answer: We have added more information in the revised version of this manuscript. Notwithstanding, due to space limitations we have not been able to provide great detail on the statistical modeling algorithm that resulted in HIFi-CCS (and the subsequently added breast cancer signature, HIFi-BCS). The statistical methods and gene ‘seeds’ are based upon Dekervel *et al.*, as the referee correctly points out. However, as the reviewer can appreciate, we have built upon these data by adding stringent analyses utilizing correlation matrices that result in a more compact, 11-gene signature, amenable to smaller-scale studies using not only RNAseq, but RT-qPCR. We believe that this is a methodological contribution with translational and practical advantages, allowing to evaluate HIF α transactivity in a variety of contexts, but is nonetheless beyond the main focus of this study. We are preparing separate manuscripts devoted to disseminate this novel methodology to the scientific community in greater detail. It would be impossible to appropriately showcase both the image quantification and transcriptomic methods in sufficient detail without compromising the translational interpretability of the HIF α - related information hereby provided.

Referee #3:

Referee #3: The manuscript "Low-dose metronomic chemotherapy offsets HIF1 α upregulation in metastatic colon cancer" by Schito et al. aims to define the differential impact of maximum tolerated dose (MTD) and low-dose metronomic (LDM) chemotherapy on spontaneous metastases arising from orthotopically implanted colon cancer fragments. The authors report organ-specific functions of HIF α paralogs, where treatment with MTD capecitabine induced the expression of HIF-1 α in primary tumors and liver metastasis and HIF-2 α in lung metastasis. Indeed, treatment with LDM capecitabine could reverse the observed induction of organ-specific expression of HIF α paralogs. Further, by reanalyzing published datasets, the authors defined an 11-gene predictive signature to stratify colon cancer patients for LDM or HIF-1 α -targeting therapies.

Employing a state-of-the-art mouse metastasis model, the manuscript reveals paralog-specific functions of HIF α during metastasis. This report presents the first side-by-side comparison of MTD and LDM therapy for metastatic colon cancer. The manuscript is conceptually at the cutting edge of translational cancer biology. It is very timely for the mechanistic understanding of the underlying differences between MTD and LDM therapies. Yet, as presented, the study falls short of supporting all of the authors' claims and it is felt some more definite experiments will be required to substantiate some of the findings. Pending satisfactory revision, this could become a very substantial and important contribution to the journal.

Answer: We appreciate the reviewer’s constructive comments, and thorough appreciation for what we concur are the main translational implications of our study. In the current, revised manuscript, we have added a second metastatic breast cancer model and breast cancer signature that further support the robustness of the data obtained from the first, colon adenocarcinoma model.

Referee #3: In further advancing the work, the authors are particularly encouraged to consider the following:

1. The findings of this manuscript present to the best of the reviewer's knowledge the first side-by-side preclinical comparison of MTD and LDM therapy on metastatic colon cancer. Even though the reviewer appreciates the use of a spontaneously-metastasizing colon cancer model in this study, it will crucial to

validate the findings in a second tumor model.

Answer: We agree with the reviewer and have performed an additional series of experiments, adding automated quantification of $\approx 5,600$ images in a preclinical model of metastatic breast cancer ($n=50$). The results further strengthen the conclusions from the first model and can be seen in Fig. 3, Fig. EV2, Appendix Fig. S6A-D, and Appendix Fig. S7, adding a total of 59 panels to the original manuscript.

Referee #3: 2. Different chemotherapy regimens did not affect the growth of primary tumors (Suppl Fig. 1A). Nevertheless, the Ki67-dependent proliferation index shows distinct patterns of tumor cell proliferation for different therapeutic regimens (Fig. 1G), especially with the vehicle and the combination of LDM-CTX and MTD-CDB exhibiting a similar score. This dichotomy between different measurement techniques should be addressed and discussed appropriately. Additionally, the authors should consider performing the analysis of the Ki67-proliferation index at an earlier time point (around 2 weeks post-therapy initiation, which will be roughly equivalent to one doubling time of tumors).

Answer: In the revised manuscript, we show that HIF-1 α inversely correlated with Ki67 proliferative indexes in both colon and breast metastatic tumors and metastases. Please see Fig. 1D (third panel) and Appendix Fig. S7, respectively. It is worth to note that even though there seem to be differences in Ki67 indexes among experimental groups (Fig. 1D rightmost panel), these differences are not statistically significant, a finding that we reproduced in the second breast cancer model (not shown).

Referee #3: 3. Concerning Fig 2E, how did the authors distinguish between parenchymatous and metastatic tissue? Could it be that there are some locally-invading colon cancer cells in the parenchymal tissue? To strengthen the conclusions, the author should consider co-staining HIF-1a and HIF-2a with a cancer cell- (or hepatocyte-) specific marker to unambiguously mark the metastatic area.

Answer: Histopathological demarcation of cancer cell containing areas is in line with our previous correlation with human vimentin staining, showing excellent agreement (Schito et al., PNAS 2012). We agree that a few single migrating cancer cells might be missed out, but this should not affect the results, due to the fact that HT29 and EMT6-CDDP tumors are not infiltrative and rather nodular or lobulated, as shown in all micrographs throughout the manuscript. Please see below the original data from Schito et al., 2012, wherein an H&E section has been demarcated, de-stained and re-stained with an anti-human vimentin antibody. In this case, these are MDA-MB-231 breast cancer cells. H&E and immunohistochemistry show a high level of agreement. We are currently working on making all these processing steps independent of the presence of an operator that is proficient in oncopathology, whilst setting the mid-term goal of publishing the techniques advanced in this manuscript as methods article to be used by oncologists and cancer biologists.

Figure for reviewers removed

Referee #3: 4. The median diameter of a metastatic nodule in the liver was 689um, whereas the lung was 211um. The authors mention on page 7 that the differential induction of HIF α -paralogs might suggest their specific roles during different stages of metastatic colonization. To validate this hypothesis, it would be crucial to judge this in (possibly) an experimental model in which lung metastatic nodules can be allowed to reach approx. 700um. This might further add a crucial explanation to the observation that the lung metastatic nodule size was reduced in the LDM-CPB treatment group but was found increased in the combination (LDM-CPB + LDM-CTX) treatment group (referring to Fig.3C).

Answer: This is a fascinating observation. The reviewer can appreciate that in the revised version, we have added a second, immunocompetent breast cancer model (EMT6-CDDP) that rapidly metastasizes to the lungs. We have sacrificed the mice before mortality ensued and managed to obtain metastatic nodules, this time in the lungs, that are within $\approx 1\%$ of the size of metastatic liver nodules from the first colonic adenocarcinoma model (Fig. 3A as compared to Fig. 2A). Importantly, at identical sizes, two different metastasis models expressed the same HIF α paralog (i.e., HIF-1 α) and responded with a decrease of the latter under LDM chemotherapy. There was no drastic decrease in nodular size, nevertheless we did observe a significant decrease in the number of metastases (Fig. 3A, right). It is worth to note that the same doublet LDM regimen used in this model and study does prolong survival as previously shown by Shaked et al. (PMID: 27569209). We believe that the second model suggests that colonization of the lung involves paralog-specific events in line with the non-redundant effects of HIF-1 α versus HIF-2 α on cancer progression.

Referee #3: 5. Stratification of metastases-bearing patients is a clinically unmet challenge and has historically been one of the major bottlenecks for testing stroma-targeting therapies. By carefully reanalyzing previously-published datasets, the authors have defined an 11-gene signature to predict the vulnerability of tumor cells towards LDM chemo- or HIF α -targeting therapies. To further strengthen their

findings, the authors should consider evaluating the expression of the established panel of genes in their preclinical spontaneous metastasis model, preferably comparing the primary tumor with the multiorgan metastases.

Answer: This is an excellent suggestion that we will strive to realize in future studies due to space constraints and the significant amount of data added in this revision: total of 167 panels analyzing $\approx 9,300$ digital images. We have nonetheless confirmed an inhibition of overall intra-metastatic angiogenesis by treating the mice in the EMT6-CDDP breast cancer model with LDM chemotherapies.

Referee #3: 6. The reviewer appreciates the stringent analyses performed by the authors to investigate metastatic nodular diameter (stratification based on the median of the control group) and the expression of HIFa-paralogs following different therapeutic regimens (Fig. 2 A-C and 3 A-C). The authors should describe the method in the main text, thereby setting a benchmark for future image analysis in similar mouse metastatic models.

Answer: We agree. We have included further details, within limits, in the Materials and Methods section. In addition, we would like to mention that we intend to publish a full methods manuscript after automatizing the most operator-dependent steps in the protocol with the hope of making these tools available to the cancer research, and possibly oncology communities at large.

Referee #3: 7. The authors need to carefully inspect the manuscript for editorial errors, especially in the legend of Fig. 3 where they describe the liver instead of lungs and on page 5, where they are describing Fig. 2A and mention LDM capecitabine instead of MTD capecitabine.

Answer: Many thanks for pointing this out. These typos have been corrected in the revised version.

Referee #3: 8. The authors need to elaborate figure legends to allow standalone interpretation of the figures.

Answer: We have striven to accommodate further details, within length restrictions for an EMM Report. All statistical analyses and abbreviations are now fully defined in the legends of the revised manuscript.

2nd Jun 2020

Dear Dr. Schito,

Thank you for the submission of your revised manuscript to EMBO Molecular Medicine. We have now received the enclosed reports from the two referees who reviewed the new version of your manuscript. As you will see, they are now overall supportive of publication. However, they still raise a few concerns that should be addressed in writing.

Furthermore, before acceptance, please address the following editorial amendments:

1) Main manuscript text:

- Please answer/correct the changes suggested by our data editors in the main manuscript file (in track changes mode). This file will be sent to you in the next couple of days. Please use this file for any further modification. Address the referees' comments directly in the manuscript, and add a point-by-point rebuttal letter.
- Please remove the highlighted text.
- Title: we appreciate the modifications to reflect the referees' concerns, but could you please simplify? (i.e. "Metronomic chemotherapy offsets maximum-tolerated dose HIF α in advanced metastatic cancers")
- Please modify the references format so as to have 10 authors before et al.
- In the material and methods, please add a separate statistics section.
- Please add a data availability section: "This study includes no data deposited in external repositories".
- Data citation: Our journal encourages inclusion of *data citations in the reference list* to directly cite datasets that were re-used and obtained from public databases. Data citations in the article text are distinct from normal bibliographical citations and should directly link to the database records from which the data can be accessed. In the main text, data citations are formatted as follows: "Data ref: Smith et al, 2001" or "Data ref: NCBI Sequence Read Archive PRJNA342805, 2017". In the Reference list, data citations must be labeled with "[DATASET]". A data reference must provide the database name, accession number/identifiers and a resolvable link to the landing page from which the data can be accessed at the end of the reference. Further instructions are available at .

2) Figures

- In the figure S6A (appendix), several panels contain a "black box" (also found in the source data). Could you please clarify the reason for these boxes?
- Thank you for providing source data. Please upload then so as to have 1 file per figure. (Source data for figure 4 should be zipped together)

3) Checklist:

Please fill in the section 1a (statistics). In section 8 (animal models), please indicate the gender, age, strain, housing and husbandry conditions, and source of the mice.

In section F (data accessibility), you do not need to fill anything as this applies only to data generated in this study.

- 4) Thank you for providing a nice synopsis picture. Could you please resize so as to have 550 px-wide? Please make sure the text remains readable.

5) Thank you for providing "The Paper Explained". Please include it in the main manuscript file, before the references.

6) As part of the EMBO Publications transparent editorial process initiative (see our Editorial at <http://embomolmed.embopress.org/content/2/9/329>), EMBO Molecular Medicine will publish online a Review Process File (RPF) to accompany accepted manuscripts.

In the event of acceptance, this file will be published in conjunction with your paper and will include the anonymous referee reports, your point-by-point response and all pertinent correspondence relating to the manuscript. Let us know whether you agree with the publication of the RPF and please confirm that you would like to REMOVE the figures before publication.

I look forward to receiving your revised manuscript.

Yours sincerely,

Lise Roth

Lise Roth, PhD
Editor
EMBO Molecular Medicine

To submit your manuscript, please follow this link:

Link Not Accessible

The system will prompt you to fill in your funding and payment information. This will allow Wiley to send you a quote for the article processing charge (APC) in case of acceptance. This quote takes into account any reduction or fee waivers that you may be eligible for. Authors do not need to pay any fees before their manuscript is accepted and transferred to our publisher.

***** Reviewer's comments *****

Referee #1 (Remarks for Author):

Overall, the authors have addressed my requests in great part. There are still two aspects that need further clarification/work:

-In Fig EV1A-B, the authors show that LDM CTX regimen compared to vehicle, reduces lung metastasis size and this is inversely correlated with the increase in HIF2a. Again, how do the authors explain these effects? The authors state that HIF2a promotes early metastatic

colonization through proliferation

and do not discuss about the relevance of HIF2a induction in lung metastasis following LDM cyclophosphamide + LDM capecitabine as a MTD regimen. The only statement the authors do is the following: Importantly, intra-metastatic nodule HIF-2 α expression increased with nodule diameter (Figure EV1C), thereby suggesting dependency upon diffusion-limited hypoxia.

- although argued with literature, it is not clear the use of a new model (EMT6) to quantify metastasis, since the authors have claimed before that HT29 cells are metastatic to the liver. Furthermore, the new experiment with EMT6 shows a trend and not significant differences.

Referee #3 (Remarks for Author):

Congratulations to the authors for solidly advance the manuscript to fully address all the major scientific concerns raised by this reviewers. There are a few minor editorial issues that the authors should address before the manuscript may be ready to go to print.

1. For every figure, the authors should mark the number of mice employed. Currently, they have mentioned the number of analyzed (lung or liver) metastatic nodules, but it is important to mention the number of mice that are the source of the metastatic nodules.

2. The number of analyzed liver metastatic nodules has been reduced from 84 (in the original manuscript) to 75 (in the revised manuscript). Accordingly, the subsequent analyses have been redone. Was this done to make the median diameter of liver metastatic nodules comparable to the median of lung metastatic nodules in the newly-added breast cancer model? The authors should explain the selection criteria and the underlying reasons for the alterations in the dataset.

The authors performed the requested editorial changes.

***** Reviewer's comments *****

Referee #1 (Remarks for Author):

Overall, the authors have addressed my requests in great part. There are still two aspects that need further clarification/work:

Rev.#1: In Fig EV1A-B, the authors show that LDM CTX regimen compared to vehicle, reduces lung metastasis size and this is inversely correlated with the increase in HIF2a. Again, how do the authors explain these effects? The authors state that HIF2a promotes early metastatic colonization through proliferation and do not discuss about the relevance of HIF2a induction in lung metastasis following LDM cyclophosphamide + LDM capecitabine as a MTD regimen. The only statement the authors do is the following: Importantly, intra-metastatic nodule HIF-2 α expression increased with nodule diameter (Figure EV1C), thereby suggesting dependency upon diffusion-limited hypoxia.

Answer: Our data show that LDM monotherapies did decrease nodular diameter in lung metastases in the HT29 colon adenocarcinoma model, as pointed out by the reviewer. It is noteworthy to mention that the correlation between lung metastatic diameter and HIF-2 α levels is observed across all conditions and is direct, not inverse, suggesting an effect correlating individual metastatic size with hypoxia and thus HIF-2 α . Since said effect of LDM chemotherapies is observed in incipient lung nodules (i.e., HT29 tumors produced large liver and small lung metastases), we hypothesized asynchronous colonization among these organs and the primary tumor; however, the precise molecular pathobiology of paralog specificity within distinct metastatic stages is beyond the scope of this manuscript. In light of this, we have restrained ourselves from further speculation. Notwithstanding, we modified the results section on LDM effects in metastases to the lungs by adding the following paragraph: *“Therefore, these observations provide in vivo*

evidence supporting HIF-2 α as a promoter of early metastatic colonization through proliferation in incipient metastatic lesions (as seen in HT29 lung metastases); by contrast, HIF-1 α could play a counterbalancing role in larger secondary tumors (as seen in HT29 liver and lung EMT6-CDDP metastases), by promoting cell-cycle arrest as a protective mechanism against chemotherapy-induced tumoral ablation, in line with previously published in vitro work (Gordan et al, 2007; Hubbi et al, 2013)."

Rev.#1: Although argued with literature, it is not clear the use of a new model (EMT6) to quantify metastasis, since the authors have claimed before that HT29 cells are metastatic to the liver. Furthermore, the new experiment with EMT6 shows a trend and not significant differences.

Answer: In agreement with the reviewers' suggestions during the previous round of revision, we performed a new series of experiments (50 mice) using an advanced metastatic model of breast cancer that readily metastasizes to the lungs (EMT6/CDDP). Crucially, we have previously reported that LDM cyclophosphamide + LDM capecitabine significantly increases survival in this model (PMID: 27569209). The EMT6/CDDP syngeneic, immunocompetent model of advanced therapy-resistant breast cancer, was supplemented with an analysis of tissue perfusion, intra-metastatic hypoxia and microvascular densities. Accordingly, and in line with suggestions from the referee, we have extensively revised the manuscript text that now includes four main figures, two expanded view figures, and seven supplementary figures, with a total of 167 panels of data generated through automated and machine-learning based analyses of a total of \approx 9,300 digital images, including \approx 5,600 captures in the second animal model alone ($n= 50$ mice). Importantly, our data show that, in a manner not dissimilar to the clinical setting, metastatic diameter or size alone does not predict survival or the overall response to adjuvant chemotherapy. Instead, our data support the notion that intra-tumoral hypoxia and HIF α expression can serve as predictors and targets to improve overall survival in cancer patients burdened with metastatic colon or breast cancers. Importantly, it is relevant to mention that LDM regimens were associated with a decreased number of metastatic nodules (i.e., nine nodules in the LDM doublet group), whereas MTD capecitabine mice presented with as many as 74 nodules in the EMT6/CDDP breast cancer model (Figure 3A, *center*). This dramatic difference in the number of nodules might explain the lack of statistically significant differences among median nodular diameters; notwithstanding, there would be no better clinical result than obtaining as many nodules of zero diameter upon a particular treatment, independently of statistical considerations.

Referee #3 (Remarks for Author):

Ref.#3: Congratulations to the authors for solidly advance the manuscript to fully address all the major scientific concerns raised by this reviewers. There are a few minor editorial issues that the authors should address before the manuscript may be ready to go to print.

Answer: We thank the reviewer for her/his positive comments and suggestions.

Ref.#3: 1. For every figure, the authors should mark the number of mice employed. Currently, they have mentioned the number of analyzed (lung or liver) metastatic nodules, but it is important to mention the number of mice that are the source of the metastatic nodules.

Answer: We have added the number of mice in square parentheses, immediately below the number of nodules in each figure, in addition to number of nodules where appropriate. The exception to this rule is Figure 1, where n indicates the number of primary tumors and hence, mice per group.

Ref.#3: 2. The number of analyzed liver metastatic nodules has been reduced from 84 (in the original manuscript) to 75 (in the revised manuscript). Accordingly, the subsequent analyses have been redone. Was this done to make the median diameter of liver metastatic nodules comparable to the median of lung metastatic nodules in the newly-added breast cancer model? The authors should explain the selection criteria and the underlying reasons for the alterations in the dataset.

Answer: There has been no deliberate attempt to change the number of nodules analyzed. This small difference is due to the fact that we have further improved and automated the metastasis image isolation algorithm, as opposed to the semi-automated method used in the first version of the manuscript. This represents an attempt to refine the image analysis methods, in order to make them completely operator-independent and thus more readily adopted by the cancer research community in the mid-term future. Consistently, we re-analyzed all HT29 and EMT6/CDDP model data with the improved algorithm, wherein small, adjacent nodules were in some cases grouped by the analysis software, thus decreasing the total number of counts, a ‘drawback’ that is overcompensated by the advantage of operator- independency. The same principle applies to HT29 metastatic lung nodules. The main criterion for tissue collection and analysis in the newly-added breast cancer model was to stop mouse follow-up at the first clinical sign of distress due to metastatic burden, to prevent unnecessary suffering and loss of statistical power. Ensuring comparable median metastatic sizes among HT29 (liver) and EMT6/CDDP (lung) nodules was indeed a conscious objective when designing experiments for the second model in order to address the reviewers’ comments; however, the degree of consistency is serendipitous. Importantly, these results have allowed us to compare almost identically sized nodules derived from metastasizing colon and breast cancers, demonstrating for the first time that HIF α paralog specificity depends upon intra-tumoral diffusion- limiting

hypoxia, wherein HIF-1 α is seen in large, established liver (HT29) or lung (EMT6/CDDP) nodules, in contrast to HIF-2 α , whose expression is less intense or absent in small, incipient metastases in both models, a finding whose universality will need to be disproved with further studies in other cancer types. Interestingly, LDM chemotherapy was able to offset HIF-1 α in large metastases, independently of nodular size, a finding with implications for paralog- specific targeting of HIF α in the clinic. We thank the reviewer for her/his suggestion of a second model, that has generated fascinating insights on the effect of metastatic size upon HIF α paralog specificity.

23rd Jun 2020

Dear Dr. Schito,

Thank you for submitting your revised version of the manuscript. I have now looked at everything and all is fine. I am therefore very pleased to accept your manuscript for publication in EMBO Molecular Medicine!

It will now be sent to our publisher to be included in the next available issue of EMBO Molecular Medicine.

Please read below for additional important information regarding your article, its publication and the production process.

Congratulations on a nice study!

With my best wishes

Lise Roth

Lise Roth, Ph.D
Editor
EMBO Molecular Medicine

Follow us on Twitter @EmboMolMed
Sign up for eTOCs at embopress.org/alertsfeeds

Corresponding Author Name: Dr. Luana Schito and Dr. Robert S. Kerbel

Manuscript Number: EMM-2019-11416